# Climate Differently Impacts the Growth of Coexisting Trees and Shrubs under Semi-Arid Mediterranean Conditions

**Jesús Julio Camarero** [1,*], **Cristina Valeriano** [1], **Antonio Gazol** [1], **Michele Colangelo** [1,2] **and Raúl Sánchez-Salguero** [1,3]

1. Instituto Pirenaico de Ecología (IPE-CSIC), 50192 Zaragoza, Spain; ecocristinavaleriano@gmail.com (C.V.); agazolbu@gmail.com (A.G.); michelecolangelo3@gmail.com (M.C.); rsanchez@upo.es (R.S.-S.)
2. School of Agricultural, Forest, Food and Environmental Sciences (SAFE), University of Basilicata, 85100 Potenza, Italy
3. Departamento de Sistemas Físicos, Químicos y Naturales, Universidad Pablo de Olavide, 41013 Sevilla, Spain
* Correspondence: jjcamarero@ipe.csic.es; Tel.: +34-976-363-222 (ext. 880041)

**Abstract:** Background and Objectives—Coexisting tree and shrub species will have to withstand more arid conditions as temperatures keep rising in the Mediterranean Basin. However, we still lack reliable assessments on how climate and drought affect the radial growth of tree and shrub species at intra- and interannual time scales under semi-arid Mediterranean conditions. Materials and Methods—We investigated the growth responses to climate of four co-occurring gymnosperms inhabiting semi-arid Mediterranean sites in northeastern Spain: two tree species (Aleppo pine, *Pinus halepensis* Mill.; Spanish juniper, *Juniperus thurifera* L.) and two shrubs (Phoenicean juniper, *Juniperus phoenicea* L.; *Ephedra nebrodensis* Tineo ex Guss.). First, we quantified the intra-annual radial-growth rates of the four species by periodically sampling wood samples during one growing season. Second, we quantified the climate–growth relationships at an interannual scale at two sites with different soil water availability by using dendrochronology. Third, we simulated growth responses to temperature and soil moisture using the forward, process-based Vaganov-Shashkin (VS-Lite) growth model to disentangle the main climatic drivers of growth. Results—The growth of all species peaked in spring to early summer (May–June). The pine and junipers grew after the dry summer, i.e., they showed a bimodal growth pattern. Prior wet winter conditions leading to high soil moisture before cambium reactivation in spring enhanced the growth of *P. halepensis* at dry sites, whereas the growth of both junipers and *Ephedra* depended more on high spring–summer soil moisture. The VS-Lite model identified these different influences of soil moisture on growth in tree and shrub species. Conclusions—Our approach (i) revealed contrasting growth dynamics of co-existing tree and shrub species under semi-arid Mediterranean conditions and (ii) provided novel insights on different responses as a function of growth habits in similar drought-prone regions.

**Keywords:** dendroecology; drought; *Ephedra nebrodensis*; *Juniperus phoenicea*; *Juniperus thurifera*; *Pinus halepensis*; radial growth; VS-Lite model



## 1. Introduction

The Mediterranean Basin is a climate warming and biodiversity hotspot, where aridification trends have been observed, which are expected to be magnified by warmer conditions during the late 21st century, negatively impacting its diverse woody flora [1]. In this region, droughts during the late 20th century and early 21st century have been among the most intense of the past millennium [2]. Such warmer and drier conditions are particularly affecting forests dominated by conifers (pines, firs, cedars, and junipers, among others), triggering dieback episodes, reducing productivity, and increasing mortality rates [3,4]. The long-term effects of climate and drought on the growth of these gymnosperm tree species have been identified through dendroecological and ecophysiological analyses [5–8], but we lack comparative analyses on coexisting gymnosperm

trees and shrubs (however, see [9]). This gap of research is very relevant because in the driest regions of the Mediterranean Basin and other semi-arid regions, treeless steppe-like landscapes are dominated by shrub species, often forming shallow roots with poor access to deep soil water, which may be sensitive to a severe winter-to-spring water deficit if they show an early growth onset [8]. Previous studies have compared radial growth in coexisting tree and shrub species inhabiting Mediterranean, dry areas [9–12], but most of them were angiosperms, except for some juniper species (*Juniperus phoenicea* L.), which showed widespread drought-induced dieback and mortality [9,12,13]. Additional research is required to compare how climate, and particularly low soil moisture water availability, influence the growth of co-occurring tree and shrub gymnosperms.

Gymnosperms from the Mediterranean Basin and other dry and semi-arid regions comprise diverse growth habits from trees (e.g., pines) to small shrubs (e.g., some junipers) [14]. This taxonomic group includes conifers (Pinophyta; e.g., Pinaceae and Cupressaceae families), but also other morphologically varied antique groups, such as Gnetophyta, where the Ephedraceae family is included. This family consists of the *Ephedra* genus, which comprises about 60 vessel-bearing species, occurring as shrubs or vines and inhabiting dry sites in tropical, subtropical, and temperate areas from the northern and southern hemispheres [15]. Furthermore, *Ephedra* have been particularly successful in colonizing dry habitats on rocky and sandy substrates thanks to adaptive trends, such as high water-use efficiency, elevated conductivity through their wide vessels, and high photosynthesis rates [16,17]. They also present plastic hydraulic traits, including shifts in wood anatomy (e.g., shifting from vessel-bearing to nearly vessel-less rings) [17,18]. Other adaptive wood-anatomical responses to prevent xylem cavitation include changes in the number of vessel groupings and the degree of helical thickening. For instance, *Ephedra* species growing in cold regions form vessels and tracheids with more pronounced helical thickenings [17,18]. Therefore, it could be expected that in *Ephedra* shrubs from dry and semi-arid regions, which form annual rings [19], radial growth is constrained by drought because wide earlywood conduits are more prone to cavitation and would lose hydraulic conductivity, thus reducing the cambial activity. To the best of our knowledge, few dendrochronological studies have been carried out with *Ephedra* species from arid and semi-arid regions (however, see [20,21]).

Regarding juniper species, they are very successful pioneer species in dry habitats because of their small lumen area, bimodal radial growth pattern, and shallow roots, which make them able to rapidly exploit superficial water pools [8,9,12]. These traits make junipers potentially better adapted to withstand drought stress than taller trees forming wider tracheids, such as pines, since junipers may experience a lower chance of a xylem embolism, whereas pines may be better able to grow during dry periods by exploiting deeper water pools.

Here we aim to compare the intra- and interannual radial growth patterns and the year-to-year growth and climate variability in four coexisting gymnosperms showing different growth habits that inhabit semi-arid Mediterranean regions: two trees (*Pinus halepensis*, *Junipers thurifera*) and two shrubs (*Juniperus phoenicea*, *Ephedra nebrodensis*). We evaluate whether the species' growth responses to climate vary as a function of site dryness by comparing dry vs. very dry sites.

## 2. Materials and Methods

### 2.1. Study Sites and the Tree and Shrub Species

The study sites were situated in the semi-arid Middle Ebro Basin, near the piedmonts of the "Sierra de Alcubierre" range, Aragón, northeastern Spain. Two mid-elevation dry sites were located in the northern (Lanaja) and southern (Monegrillo) sides of this small range situated within a steppe landscape (see Figure 1), whilst the other low-elevation, very dry site (Peñaflor) was located near the inner Middle Ebro Basin (Table 1). The Monegrillo site was located on the south-oriented steep slopes of the "Sierra de Alcubierre" range, whereas the Lanaja site was located on gentle slopes of the north-oriented, wet side of this range. Both sites present slightly different climatic conditions and vegetation due to

differences in elevation and orientation. All sites have basic soils formed by marls and gypsum, which were more abundant in the very dry site.

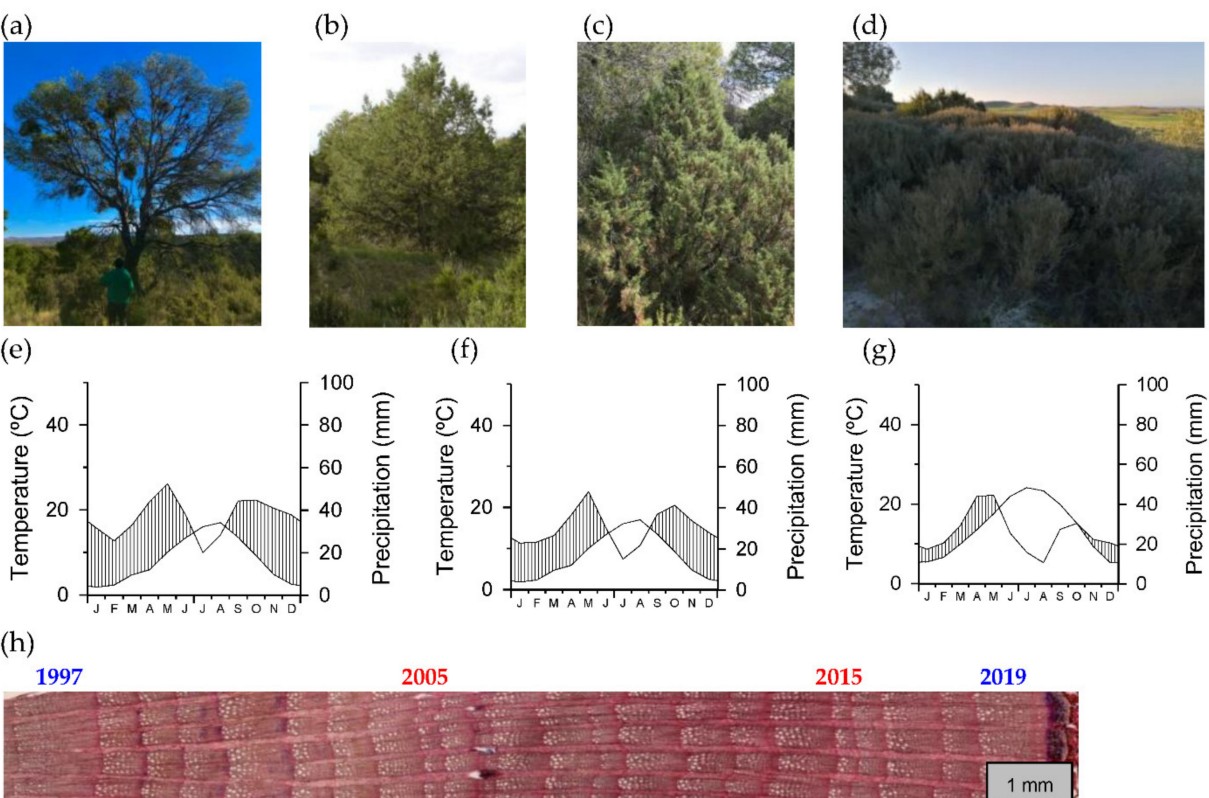

**Figure 1.** Views of the study species ((**a**) *Pinus halepensis*, (**b**) *Juniperus thurifera*, (**c**) *Juniperus phoenicea*, and (**d**) *Ephedra nebrodensis*); climate diagrams of Lanaja (**e**), Monegrillo (**f**), and Peñaflor (**g**) meteorological stations; (**h**) cross-section of *Ephedra* wood sampled from an individual located in the Lanaja site. The blue and red numbers indicate wide and narrow annual rings, respectively, situated along the cross-section, which corresponded to wet and dry conditions, respectively. Note the vessels with the small lumen area in the very wide 1997 ring.

**Table 1.** Climate characteristics of the three study sites and sampled tree and shrub species. See Figure 1 for additional climatic information.

| Site Type | Site | Latitude (N) | Longitude (W) | Elevation (m a.s.l.) | Aspect | Sampled Species | Mean Annual Temperature (°C) | Total Annual Precipitation (mm) |
|---|---|---|---|---|---|---|---|---|
| Mid-elevation dry sites | Lanaja | 41°44′16″ | 0°23′31″ | 483 | N–NE | *Pinus halepensis, Juniperus thurifera, Juniperus phoenicea, Ephedra nebrodensis* | 8.5 | 463 |
| | Monegrillo | 41°38′50″ | 0°21′50″ | 540 | E–NE | *Ephedra nebrodensis* | 8.4 | 394 |
| Low-elevation very dry site | Peñaflor | 41°47′01″ | 0°43′19″ | 358 | E, S | *Pinus halepensis, Juniperus thurifera, Juniperus phoenicea, Ephedra nebrodensis* | 14.4 | 353 |

The study sites experienced continental Mediterranean climate conditions characterized by cold winters, dry summers, and relatively wet conditions in spring and autumn (Figure 1). The vegetation was dominated by *Pinus halepensis* Mill. (Figure 1a) stands with scattered junipers (*Juniperus thurifera* L. (Figure 1b), *Juniperus phoenicea* L. (Figure 1c), and *Juniperus oxycedrus* L.), some evergreen oaks (*Quercus coccifera* L., *Quercus ilex* L.), and shrubs (*Rhamnus lycioides* L., *Salvia rosmarinus* (L.) Sheild., *Genista scorpius* (L.) DC., *Globularia alypum* L., *Lynum sufruticosum* L., and *Thymus* spp.). These shrub species are angiosperms, except for *J. phoenicea*, but one major gymnosperm shrub was also present in

this dry maquis-type shrubland, *Ephedra nebrodensis* Tineo ex Guss. (hereafter referred to as *Ephedra* (Figure 1d)).

According to data from nearby climatic stations (Lanaja, 0°19′44″ W, 41°46′19″ N, 369 m a.s.l.; Monegrillo, 0°24′57″ W, 41°38′19″ N, 432 m a.s.l.; Zaragoza-Aula Dei, 0°48′37″ W, 41°43′54″ N, 231 m a.s.l.), the annual precipitation ranged between 463 (Lanaja dry site) to 353 mm (Peñaflor very dry site). The mean annual temperature ranged between 8.4 °C (Monegrillo dry site) to 13.8 °C (Peñaflor very dry site) (Table 1). The mean minimum and maximum temperatures ranged from −9.2 to 38 °C. The annual climatic water balance (the difference between the precipitation and potential evapotranspiration) ranged from −410 mm (Lanaja dry site) to −620 mm (Peñaflor very dry site). The water balance was negative from April to September and reached the minimum values in July and August. The relative air humidity varied from 33 to 97%. Frosts can occur from November until March with a mean frequency of 13–26 days per year. Radiation fogs linked to high pressures are also frequent in the Ebro Basin from November to February (during December and January, the mean frequency of foggy days is 24% per month), and the fog layer is usually 300 to 350 m thick [22].

### 2.2. Field Sampling

Field sampling was carried out during January (*Ephedra*) and October (rest of the species) in 2020. We sampled dominant, apparently healthy individuals (15–35 individuals per species) of the four study species. We measured their basal diameter and total height using tapes and a laser rangefinder (Nikon Forestry Pro II, Tokyo, Japan), respectively. In the case of trees, the diameter was measured at 1.3 m. In *Ephedra*, the height was measured with tapes.

We took two cores per individual at 1.3 m using Pressler increment borers in all species, except for *Ephedra*, in which case, we took basal cross-sections using a handsaw. In *J. phoenicea*, we took cross-sections at 0.5–1.0 m of 5–10 individuals per site to facilitate the cross-dating of cores.

### 2.3. Xylem Formation: Intra-Annual Growth Rates

In the very dry site (Peñaflor) we characterized the intra-annual growth rates during 2010 (*Ephedra*) and 2020 (the other three species) by periodically taking wood samples of five individuals per species. The 2010 and 2020 study years were relatively dry (2010, mean annual temperature 15.3 °C, annual precipitation 267 mm) and wet (2020, mean annual temperature 16.5 °C, annual precipitation 391 mm), respectively. We took small shoots of diameter 2–6 mm in *Ephedra* and microcores in the other species (extracted using a Trephor® microcorer (Belluno, Italy) every 14–30 days. Then, we obtained transversal sections (15–20 μm thick) using a sliding microtome (Leica SM2010 R, Leica Biosystems, Nussloch, Germany). Sections were mounted on glass slides, stained with 0.05% cresyl violet acetate, and fixed with Eukitt® (Orsatec, Bobingen, Germany) to quantify the amount of newly formed xylems (not lignified cells) which were stained as blue tissue. Images of sections were taken at 40–100x magnification with a digital camera mounted on a light microscope (Olympus BH2, Tokyo, Japan), and then we measured the width of the new xylems to obtain radial growth rates using the ImageJ analysis software (ver. 1.5i, NIH, Bethesda, MD, USA).

### 2.4. Climate Data

We obtained long (period 1950–2020), homogeneous series of monthly climate data (mean temperature, total precipitation) from the Climate Explorer webpage [23] by considering the 0.1° grid located over each study site. These monthly climate data were transformed into seasonal data by averaging (temperature) or summing (precipitation) the monthly data.

To compare the climate–growth relationships at different time resolutions, we also obtained weekly climate data for the period 1961–2019 from a high-resolution (1.1 km²) Span-

ish dataset [24]. Specifically, for each site, we obtained a series of mean maximum (TMx) and minimum (TMn) temperatures, climatic water balance (P-PET, difference between precipitation and reference evapotranspiration calculated using the FAO56 Penman–Monteith equation), relative air humidity (RH), and vapor pressure deficit (VPD).

Finally, we obtained estimates of the soil moisture (corresponding to the upper 10 cm), gridded at 1.00–1.25° for the period 1979–2016, corresponding to remote-sensing products of the Climate Change Initiative (CCI) dataset of the European Space Agency (ESA) [25].

### 2.5. Dendrochronological Data

The wood samples were processed to calculate the year-to-year radial growth variability, quantified via the ring width, by using dendrochronological methods [26]. Samples were air-dried and sanded with sandpapers of progressively finer grain until the annual rings were clearly visible. Then, samples were visually cross-dated by annotating narrow rings and the cross-dating was verified using COFECHA software (ver. 6.06P, Laboratory of Tree-Ring Research, The Univ. of Arizona, AZ, USA) [27]. In all cases, two radii per individual tree or shrub were dated, and the ring widths were measured along them to the nearest 0.01 mm using a Lintab-TSAP measuring device (Rinntech$^{TM}$, Heidelberg, Germany). In the cross-sections, we measured the whole radius from the youngest ring near the bark to the oldest ring near the pith.

The resulting tree ring width data were detrended and standardized using the AR-STAN ver.44 software (Tree-Ring Lab, Columbia University, NY, USA) [28]. Detrending allowed for removing growth trends due to changes in size, age, and stand dynamics. The detrending was done by fitting a 67% cubic smoothing spline with a 50% cutoff frequency [26]. Then, the resulting detrended series were pre-whitened with low-order autoregressive models to remove the growth persistence. Individual, pre-whitened series of ring width indices (RWI) were combined into site mean residual series or chronologies using bi-weight robust means [26].

Several tree ring statistics were calculated for each chronology (see Table 2): mean ring width and its standard deviation (SD); the first-order autocorrelation (AC1) of the ring width data, which quantified the serial dependence between rings; the mean sensitivity (MS) of standard ring width indices, which measured the relative change in width between consecutive rings; the mean correlation (Rbar) between individual indexed series, which accounted for the coherence in growth variability within each species in each site [29]. Lastly, we defined the best-replicated period of each chronology by calculating its expressed population signal (EPS), which measures how well replicated the chronology was, considering a minimum threshold of EPS ≥ 0.85 [30].

**Table 2.** Size and number of processed individuals and corresponding ring width statistics. Values are means ± standard deviations. Abbreviations: AC1, first-order autocorrelation; MS, mean sensitivity; Rbar, mean correlation between indexed ring-width series. The best replicated period was defined as that with an expressed population signal ≥ 0.85.

| Site Type | Species | Diameter (cm) | Height (m) | No. of Measured Individuals (No. of Radii) | Best Replicated Period | Mean Ring Width (mm) | AC1 | MS | Rbar |
|---|---|---|---|---|---|---|---|---|---|
| Dry site | *Pinus halepensis* | 28.7 ± 2.3 | 6.9 ± 1.3 | 32 (47) | 1939–2020 | 1.27 ± 0.23 | 0.72 | 0.36 | 0.78 |
| | *Juniperus thurifera* | 17.0 ± 1.8 | 5.8 ± 0.4 | 16 (28) | 1957–2020 | 1.19 ± 0.69 | 0.41 | 0.43 | 0.52 |
| | *Juniperus phoenicea* | 11.2 ± 1.6 | 1.9 ± 0.5 | 17 (31) | 1943–2020 | 0.56 ± 0.27 | 0.46 | 0.39 | 0.58 |
| | *Ephedra nebrodensis* | 10.0 ± 1.8 | 1.1 ± 0.1 | 20 (40) | 1989–2019 | 0.73 ± 0.30 | 0.41 | 0.35 | 0.34 |
| Very dry site | *Pinus halepensis* | 32.3 ± 4.6 | 7.8 ± 1.8 | 23 (46) | 1917–2020 | 0.99 ± 0.66 | 0.64 | 0.47 | 0.80 |
| | *Juniperus thurifera* | 17.5 ± 2.9 | 6.1 ± 0.6 | 21 (42) | 1941–2020 | 1.21 ± 0.87 | 0.62 | 0.41 | 0.60 |
| | *Juniperus phoenicea* | 11.7 ± 1.5 | 1.8 ± 0.2 | 14 (28) | 1934–2020 | 0.51 ± 0.24 | 0.32 | 0.51 | 0.61 |
| | *Ephedra nebrodensis* | 8.7 ± 0.7 | 0.9 ± 0.1 | 20 (39) | 1985–2019 | 0.44 ± 0.21 | 0.45 | 0.36 | 0.38 |

*2.6. Statistical Analyses*

We checked the normality of the variables by using Shapiro–Wilk tests. To compare variables between sites we used Student's *t*-tests. Pearson correlations were calculated between seasonal (mean temperatures, summed precipitation), monthly, and weekly climate variables and the species' chronologies for the common period of 1961–2019, except in the case of the shorter *Ephedra* series, which encompassed the period of 1989–2019. In the case of seasonal and monthly climate variables, correlations were calculated from the prior to the current September. In all cases, we considered the 0.05 and 0.01 significance levels.

Linear mixed effect models [31] were used to test for the differences in the ring width characteristics between sites (i.e., dry vs. very dry) and between growth forms (shrub vs. tree). Particularly, we compared the ring width, its autocorrelation between series of shrubs and trees, and between series in dry and very dry sites. To compare ring width characteristics between shrubs and trees, the chronology identity was used as a random factor to account for the fact that series were gathered in different sites. To compare the ring width characteristics between dry and wet places, species identity was used as a random factor to account for the fact that four different species were studied at the two sites. In the two cases, a constant variance structure was included to account for the fact that the number of series (and thus, the variance) varied between chronologies. Models were fitted using the nlme package [32] in the R statistical environment ve. 4.0.3 (The R Foundation, Vienna, Austria) [33].

We also used linear mixed effect models to test for the relationship between ring width and drought severity. The 12-month-long June Standardized Precipitation Evapotranspiration Index (SPEI) was used to assess the drought severity following previous studies [7]. We compared the responses of residual, pre-whitened series of the ring width indices to drought severity and how it varied between sites (i.e., dry vs. very dry sites) and growth form (shrub vs. tree). The same random structure was used as explained before. We proposed models including the SPEI and its interaction with site type or growth form for the period 1989–2019. Fitted models were ranked according to their Akaike information criterion (AIC) and the most parsimonious model showing the lowest AIC was selected.

*2.7. VS-Lite Growth Model*

We used the VS-Lite forward growth model to assess differences in the climatic controls of tree growth between sites for the four species. The VS-Lite forward model was used to characterize the climatic drivers of radial growth [34–36]. The model formulation contains several parameters: a growth–temperature parameter (gT) and its two subparameters determining the minimum ($T_1$) and optimal ($T_2$) growth temperatures, and the growth–soil moisture parameter (gM) and its two subparameters determining the minimum ($M_1$) and optimal ($M_2$) soil moisture [33]. The VS-Lite simulates nonlinear growth response of the mean series of ring width indices as a function of monthly temperature and precipitation, based on the principle of limiting factors [34]. To estimate the model parameters, we followed a Bayesian framework [35]. The $T_1$ and $M_1$ subparameters determine when growth will occur, and the $T_2$ and $M_2$ subparameters determine when growth is not limited anymore by temperature and soil moisture, respectively. We estimated the solar radiation (gE parameter) from the site latitude by considering no interannual variability. We used these parameters to simulate the ring width indices for the 1960–2019 calibration period; then, we divided this period in two subperiods to evaluate the temporal stability of the growth responses, except in the case of the short *Ephedra* series. We assumed uniform priors for the growth function parameters, and independent, normally distributed errors for the ring width indices. Then, 10,000 iterations were run using three parallel chains and a white Gaussian noise model error [35,36]. Snow dynamics were not considered since snowfall is rare at the study sites. To estimate monthly soil moisture from temperature and total precipitation, the model used the empirical leaky bucket model of hydrology, whilst other parameters (e.g., runoff, root depth, and growing season length) were taken from previous studies on similar sites and species [13,37].

## 3. Results

### 3.1. Intra-Annual Growth Patterns

Radial growth started during late March to April and ended from November to December (Figure 2). *Ephedra* started growing rapidly in April to May, before the pine and junipers. Maximum radial growth rates were observed in May and early June (0.01–0.02 mm day$^{-1}$; 0.008 mm day$^{-1}$ in *Ephedra*), followed by drops in growth rates from July to September, and a second peak in September and October (0.003–0.007 mm day$^{-1}$), which was more noticeable for the *P. halepensis* and the two juniper species but not for the *Ephedra* (Figure 2). The second growth peak for the *J. phoenicea* was less intense and delayed with respect to those of the *P. halepensis* and *J. thurifera*.

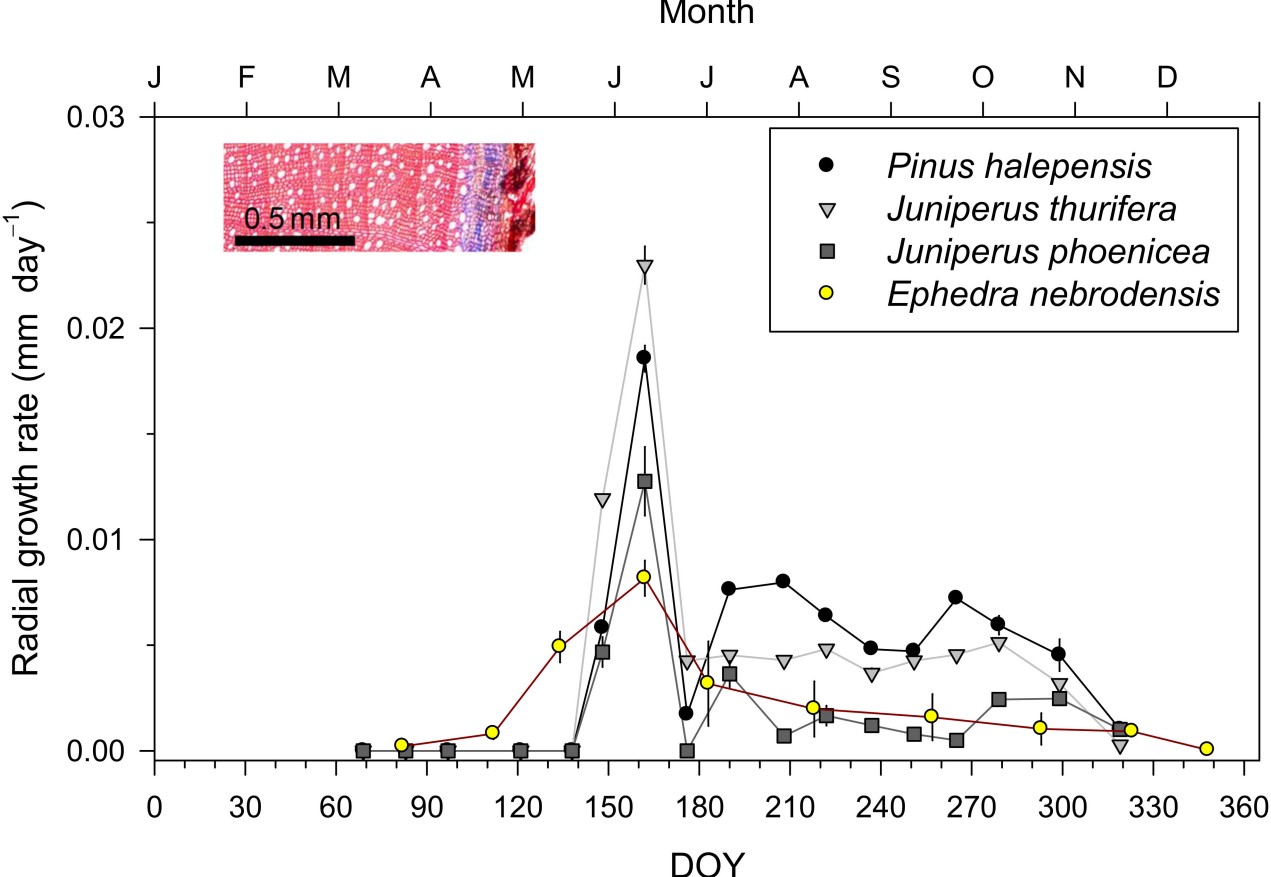

**Figure 2.** The intra-annual radial growth rates were calculated for the four study species in the very dry site. Values are means ± standard errors. The right *y*-axis corresponds to *Ephedra* rates. The image shows a stained *Ephedra* cross-section sampled in May showing recently formed xylem as blue cells. DOY: day of the year.

### 3.2. Tree Ring Width Series and Statistics

The maximum ages of the sampled *Ephedra* individuals were 84 and 72 years in the dry and very dry sites, respectively, but the best-replicated period in this species was much shorter (31–35 years, common period 1989–2019) than in the others (64–87 years, common period 1957–2020) (Table 2). We found significantly lower ring width values ($t$ = 6.00, $p$ < 0.01) and first-order autocorrelation ($t$ = 2.92, $p$ = 0.027) in shrubs than in trees (Figure S1). We also found significantly lower ring width values ($t$ = 5.37, $p$ < 0.01) and first-order autocorrelation ($t$ = 3.36, $p$ < 0.01) in very dry sites than in dry sites. Regarding the mean sensitivity, it was similar (0.41) in trees and shrubs ($t$ = 0.35, $p$ = 0.76). The coherence between individual series (Rbar) was higher in the very dry site (0.60) than in the dry site

(0.55), but the difference was not significant ($t = 0.34$, $p = 0.66$). The Rbar was higher in trees (0.67) than in shrubs (0.48), but the difference was not significant ($t = 2.04$, $p = 0.13$).

The growth rates showed high and low values corresponding to wet and dry years, respectively, such as 1997 (highest decile of annual precipitation) and 2005 (lowest decile of annual precipitation) (Figure 3). Other recent dry years with low ring width index values were 1995, 2012, and 2019.

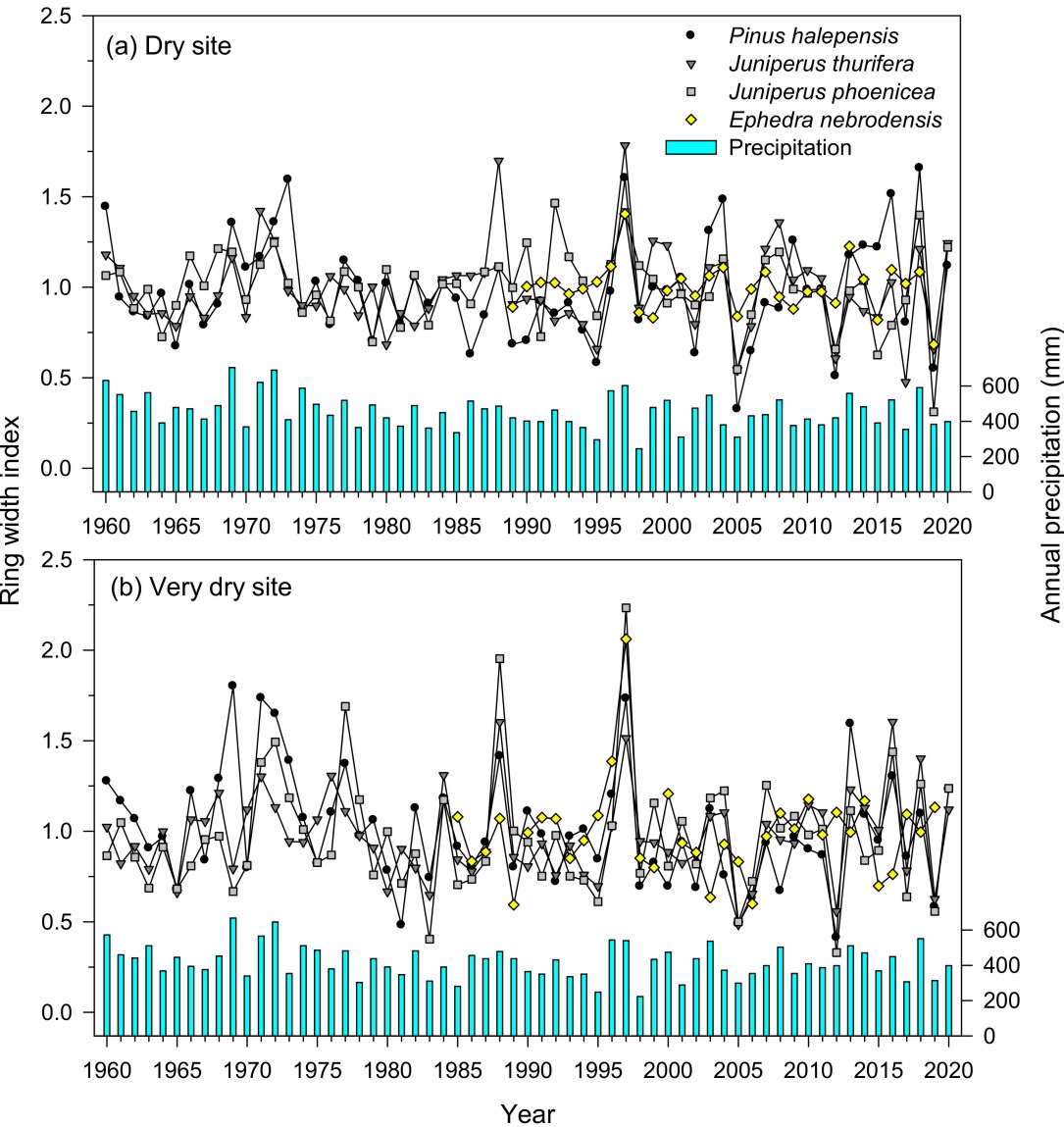

**Figure 3.** The interannual radial growth rates (ring width indices, residual chronologies) that were calculated for the four study species at the two study sites. The bars show the annual precipitation (right *y*-axes).

Considering the common period (1989–2019), in the dry site, all species showed significant ($p < 0.01$) correlations between their mean series or chronologies, with the highest ($r = 0.67$) and lowest ($r = 0.45$) Pearson coefficients corresponding to the *P. halepensis–J. thurifera* and *P. halepensis–J. phoenicea* pairs, respectively. In the very dry site, all species' chronologies showed significant ($p < 0.03$) correlations, except for the pair *Ephedra–J. thurifera* ($r = 0.28$, $p = 0.13$). In this site, both junipers' chronologies presented the highest correlation value ($r = 0.82$), followed by the *P. halepensis–J. thurifera* ($r = 0.77$) and *P. halepensis–J. phoenicea* ($r = 0.74$) pairs. *Ephedra*'s chronologies showed the highest correlations with

*P. halepensis* or *J. phoenicea* chronologies in either the dry site (*r* = 0.58) or the very dry site (*r* = 0.41).

In the comparison between sites, all species' chronologies showed significant (*p* < 0.01) and positive correlations. Trees showed higher between-site associations (*P. halepensis*, *r* = 0.68; *J. thurifera*, *r* = 0.70) than shrubs (*J. phoenicea*, *r* = 0.59; *Ephedra*, *r* = 0.52), indicating higher site-to-site variability in the growth of shrubs.

### 3.3. Climate–Growth Relationships in Trees and Shrub Species at the Monthly Scale

When considering the whole individual variability, no differences in the growth response to the 12-month June SPEI were found between shrubs and trees or between dry and very dry sites (Table S1). The analyses showed a similar impact of SPEI on the growth rates (RWI, ring width indices) in both cases (*t* = 5.31, *p* < 0.01), indicating that the growth responses to drought were significant but did not vary significantly between shrubs and trees or between dry and very dry sites. However, the analyses based on the mean site or species series showed different patterns in climate–growth associations. Overall, prior winter and spring precipitation were the climate variables that were most related to growth in the very dry and dry sites, respectively (Figures 4–6). The correlations with monthly climate data showed that the April and June wet conditions improved the *Ephedra* growth in the dry and very dry sites, respectively (Figure 4 and Figure S2). In the very dry site, growth increased as precipitation from December to January did, followed by warm February conditions, i.e., from two to three months before growth started (Figure 2 and Figure S3). At this site, dry and warm summer conditions were associated with low growth rates, which agreed with the analyses based on soil moisture. Growth increased in response to wet soil conditions in the prior winter and current summer, particularly for July. At the dry site, the same result was found for summer soil moisture.

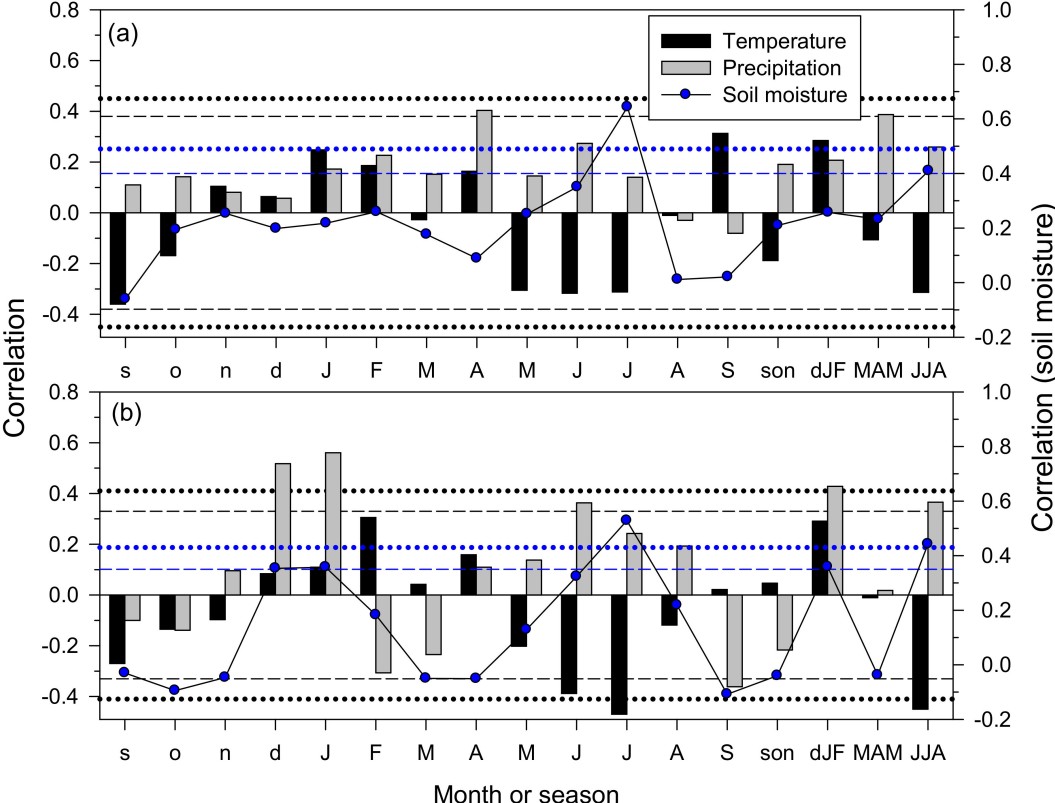

**Figure 4.** Monthly and seasonal climate–growth relationships of the *Ephedra nebrodensis* quantified at the (**a**) dry and (**b**) very dry sites. Horizontal dashed and dotted lines indicate the 0.05 and 0.01 significance levels, respectively. Note the different scales for the soil moisture (blue symbols with lines, right *y*-axes) and their significance levels (blue dashed and dotted lines). Months of the prior and current years are abbreviated using lowercase and uppercase letters, respectively.

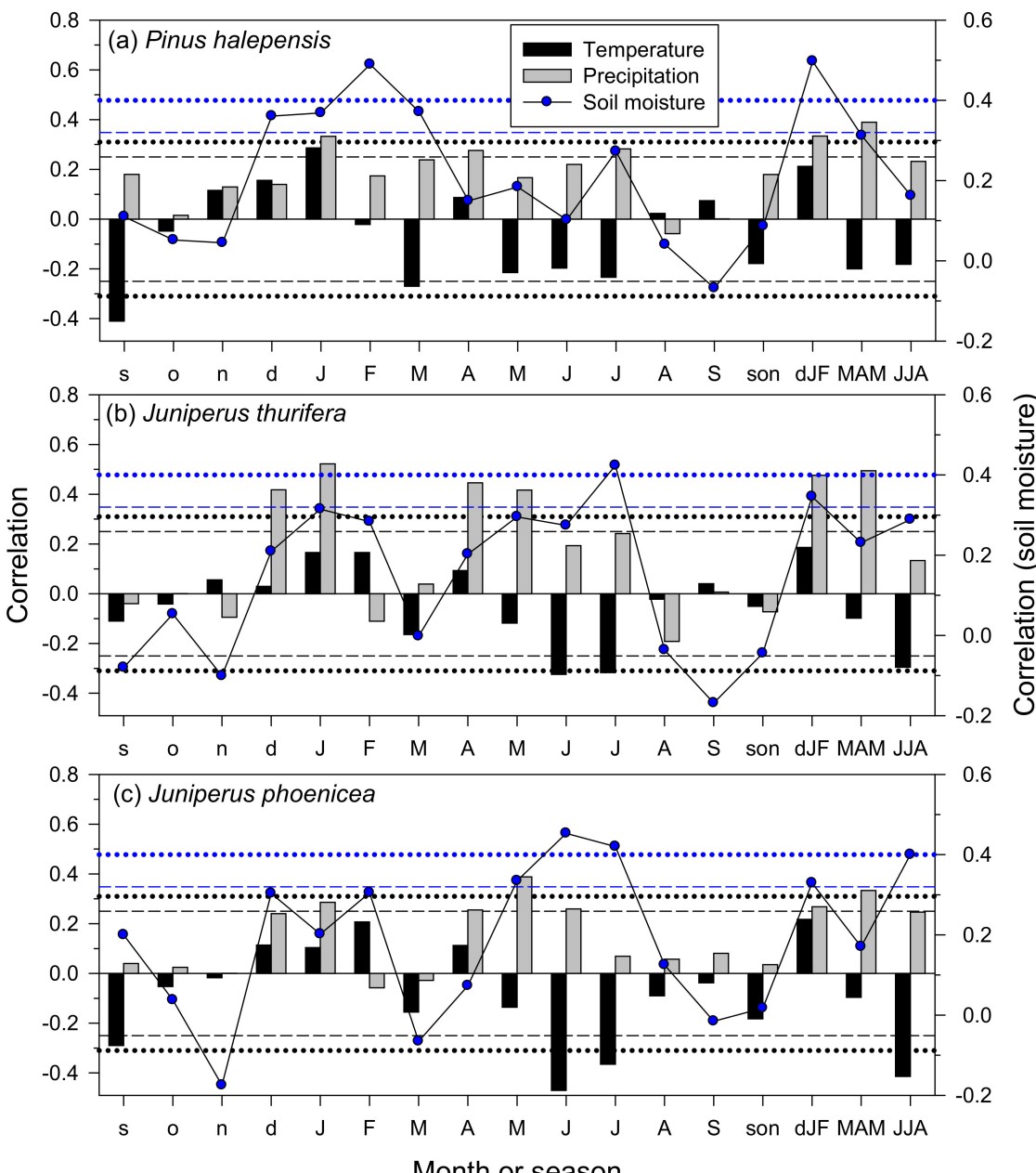

**Figure 5.** Monthly and seasonal climate–growth relationships of three species studied in the dry site. Horizontal dashed and dotted lines indicate the 0.05 and 0.01 significance levels, respectively. Note the different scales for the soil moisture (blue symbols with lines, right *y*-axes) and their significance levels (blue dashed and dotted lines). Months of the prior and current years are abbreviated using lowercase and uppercase letters, respectively.

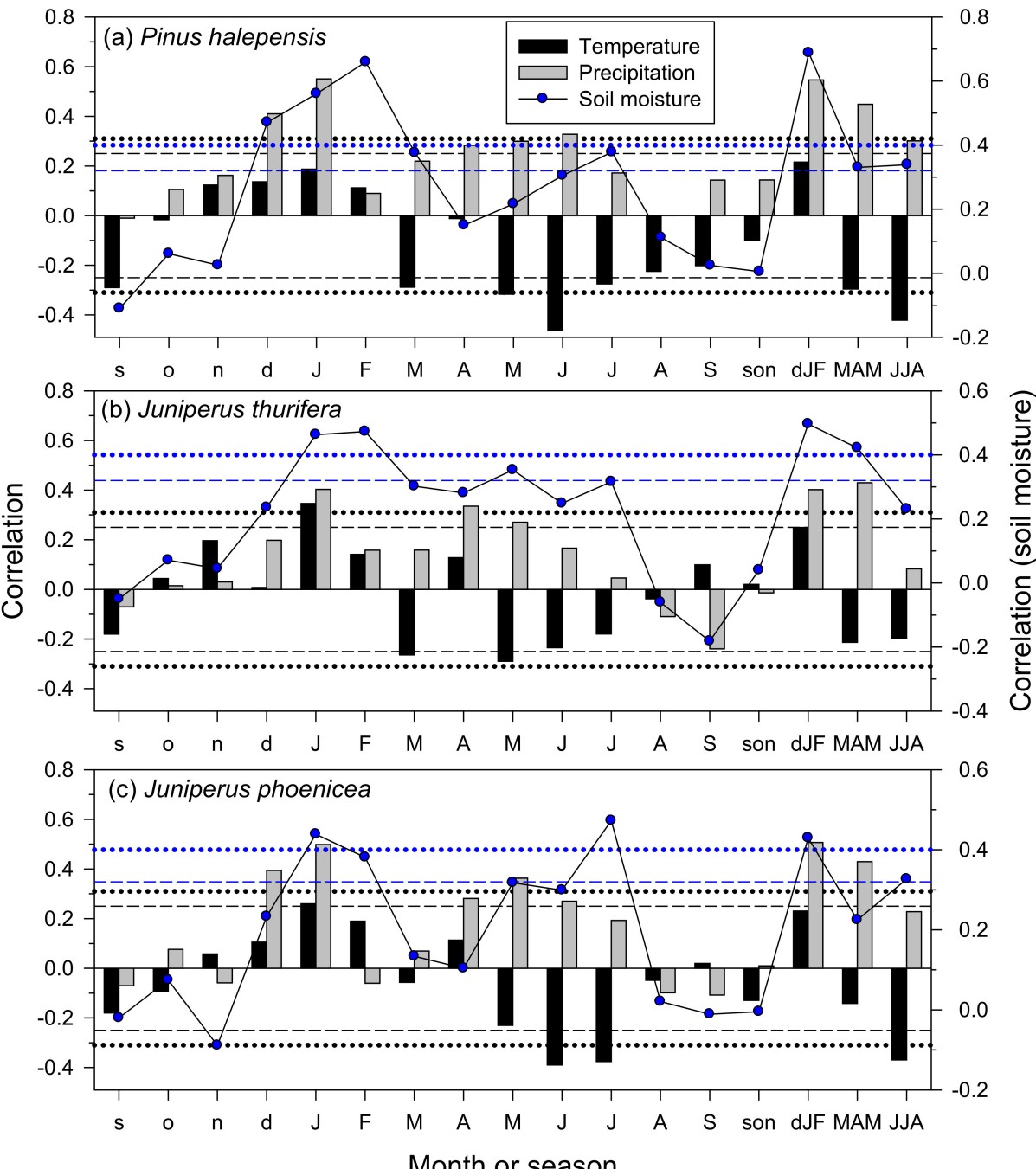

**Figure 6.** Monthly and seasonal climate–growth relationships of three species studied at the very dry site. Horizontal dashed and dotted lines indicate the 0.05 and 0.01 significance levels, respectively. Note the different scales for the soil moisture (blue symbols with lines, right *y*-axes) and their significance levels (blue dashed and dotted lines). Months of the prior and current years are abbreviated using lowercase and uppercase letters, respectively.

At the dry site, wet prior winter (December, January) and spring-to-early-summer (April to June) conditions were associated with higher growth rates of the two juniper species (Figure 5). Low June–July temperatures were related to improved growth, probably reflecting lower evapotranspiration rates. The positive relationship between wet winter-to-spring conditions and growth was also observed in this site in the case of the *P. halepensis*. In this species, high precipitation levels in January, April, and July were related to higher growth, whilst warm March conditions were related to lower growth. For the *P. halepensis* and *J. phoenicea*, warm prior September conditions, indicative of late summer drought stress, were associated with reduced growth. High soil moisture values in winter and June–July were associated with higher growth in junipers, whereas the *P. halepensis* growth was only associated with winter soil moisture.

At the very dry site, high precipitation levels in the prior winter and current spring (junipers) to early summer (June, *P. halepensis*) were related to enhanced growth (Figure 6). Cool March and May–June conditions were associated with higher growth in the two tree species, whilst the *J. phoenicea* growth was reduced by warm summer conditions. Warm prior September conditions were also related to low growth rates for the *P. halepensis*. High soil moisture levels in prior winter, spring, and summer seasons were related to high growth rates. However, and as in the dry site, the *P. halepensis* growth responded to winter soil moisture changes, and the *J. thurifera* growth responded to the soil moisture in winter and summer. At this site, the *J. phoenicea* growth also responded to winter and summer soil moisture, but with a pronounced response to summer soil moisture conditions, as observed at the dry site.

### 3.4. Climate–Growth Relationships in Trees and Shrub Species at Weekly Scales

The *Ephedra* growth rates increased as the values of the P-PET and RH did in late June and early July, and decreased as the temperature and VPD increased, particularly at the very dry site (Figure 7). Warm late September–October conditions at the dry site were also associated with higher growth rates.

For the *P. halepensis*, the growth was constrained by warm conditions, high VPD, and low RH from April to June, especially at the very dry site (Figure 8). Wet (high P-PET) January conditions also improved growth at the very dry site.

For *J. thurifera*, the growth was constrained by warm conditions from March to July, except in April, particularly at the very dry site (Figure 9). High VPD and low P-PET and RH from April to July were associated with reduced growth. Again, wet January conditions were related to high growth rates.

For *J. phoenicea*, the growth was constrained by warm June conditions, especially at the very dry site (Figure 10). Cool and wet January conditions improved the growth. High P-PET and RH values and low VPD values from May to July were related to higher growth rates, showing again the higher responsiveness at the very dry site.

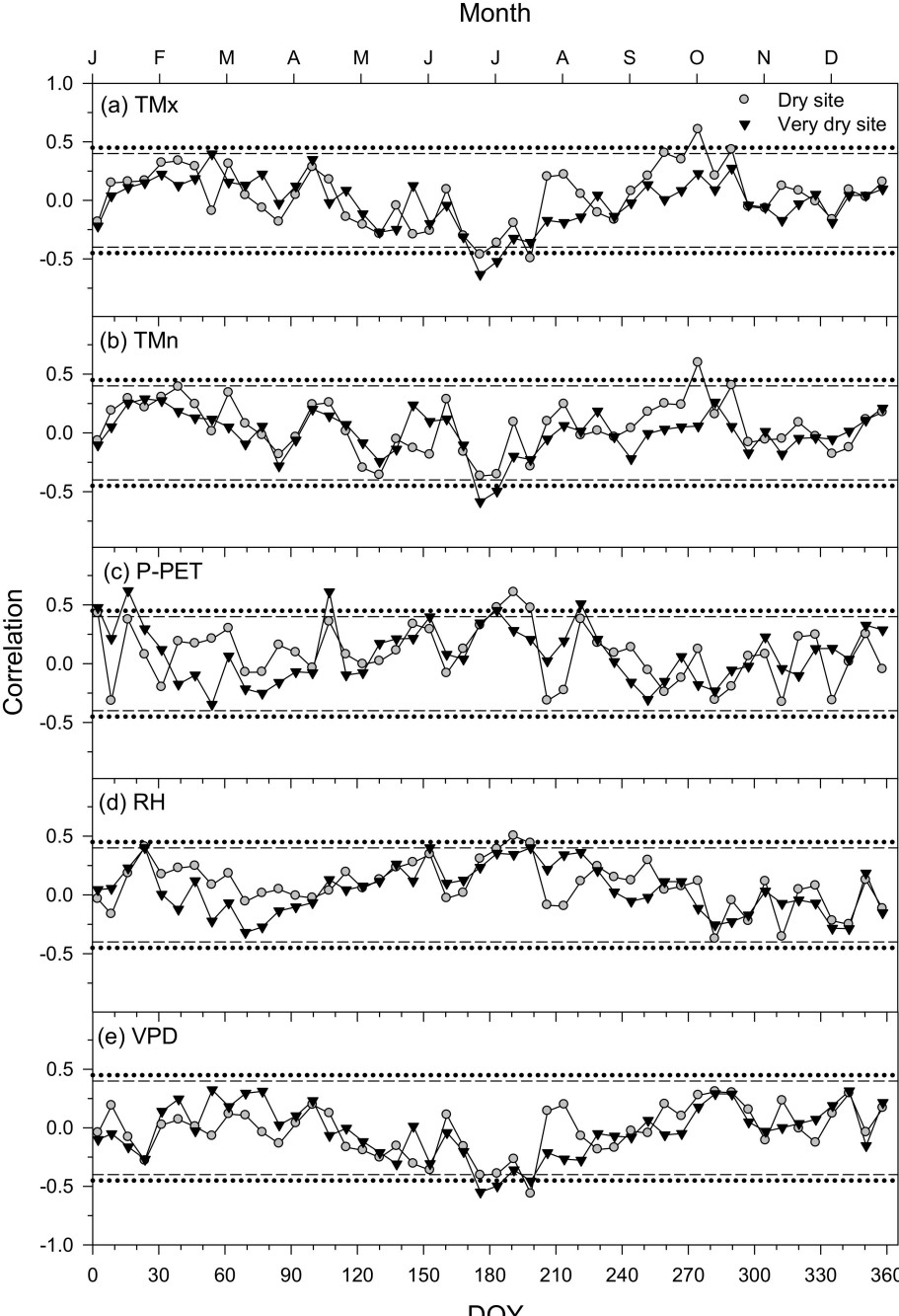

**Figure 7.** The climate–growth relationships of *Ephedra nebrodensis* were assessed at a weekly resolution at the dry and very dry sites. The analyzed climate variables were (**a**) TMx—mean maximum temperature, (**b**), TMn—mean minimum temperature, (**c**) P-PET—climatic water balance, (**d**) RH—relative humidity, and (**e**) VPD—vapor pressure deficit. The horizontal dashed and dotted lines indicate the 0.05 and 0.01 significance levels, respectively.

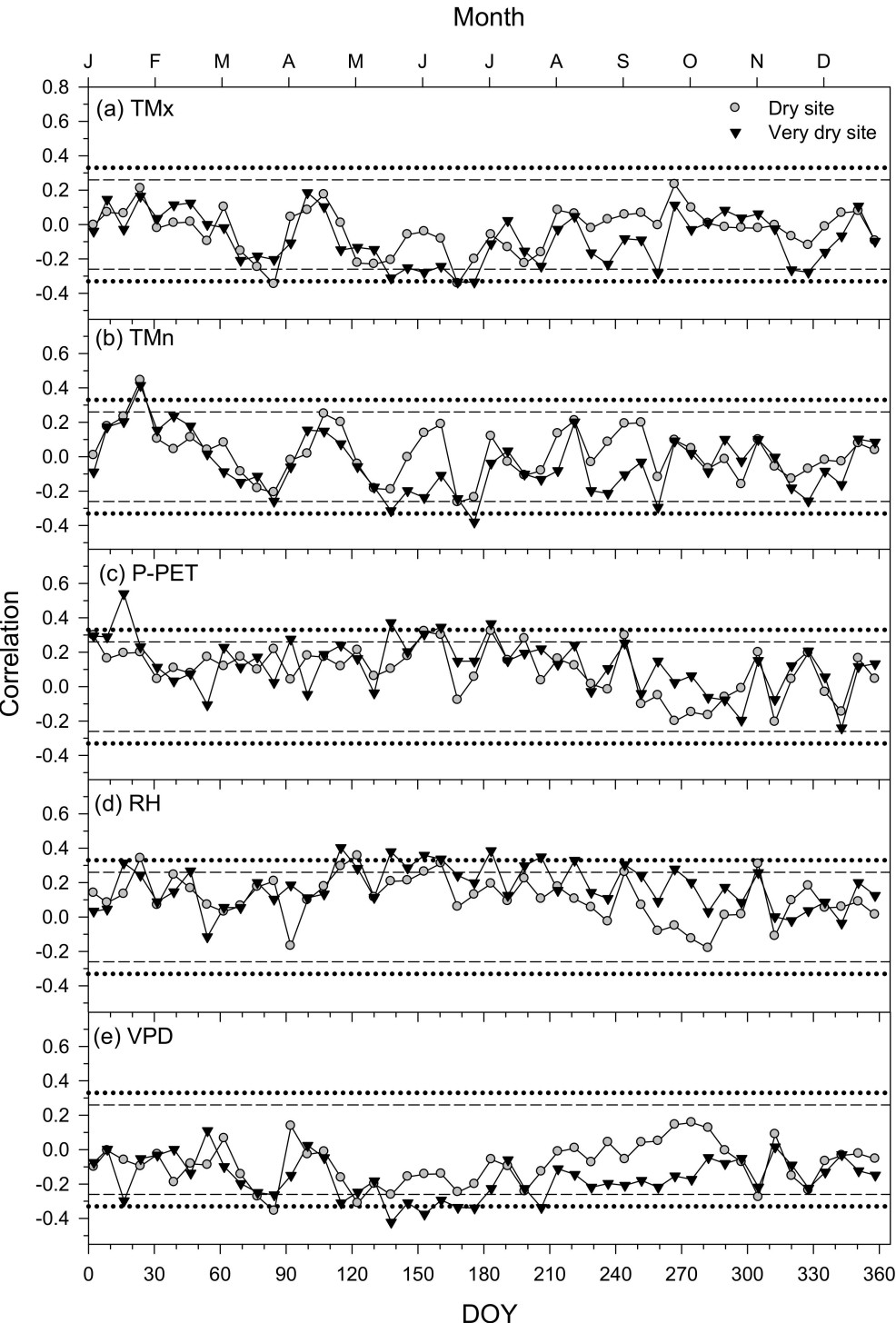

**Figure 8.** The climate–growth relationships of *Pinus halepensis* were assessed at a weekly resolution at the dry and very dry sites. The analyzed climate variables were (**a**) TMx, (**b**), TMn, (**c**), P-PET, (**d**) RH, and (**e**) VPD. The horizontal dashed and dotted lines indicate the 0.05 and 0.01 significance levels, respectively.

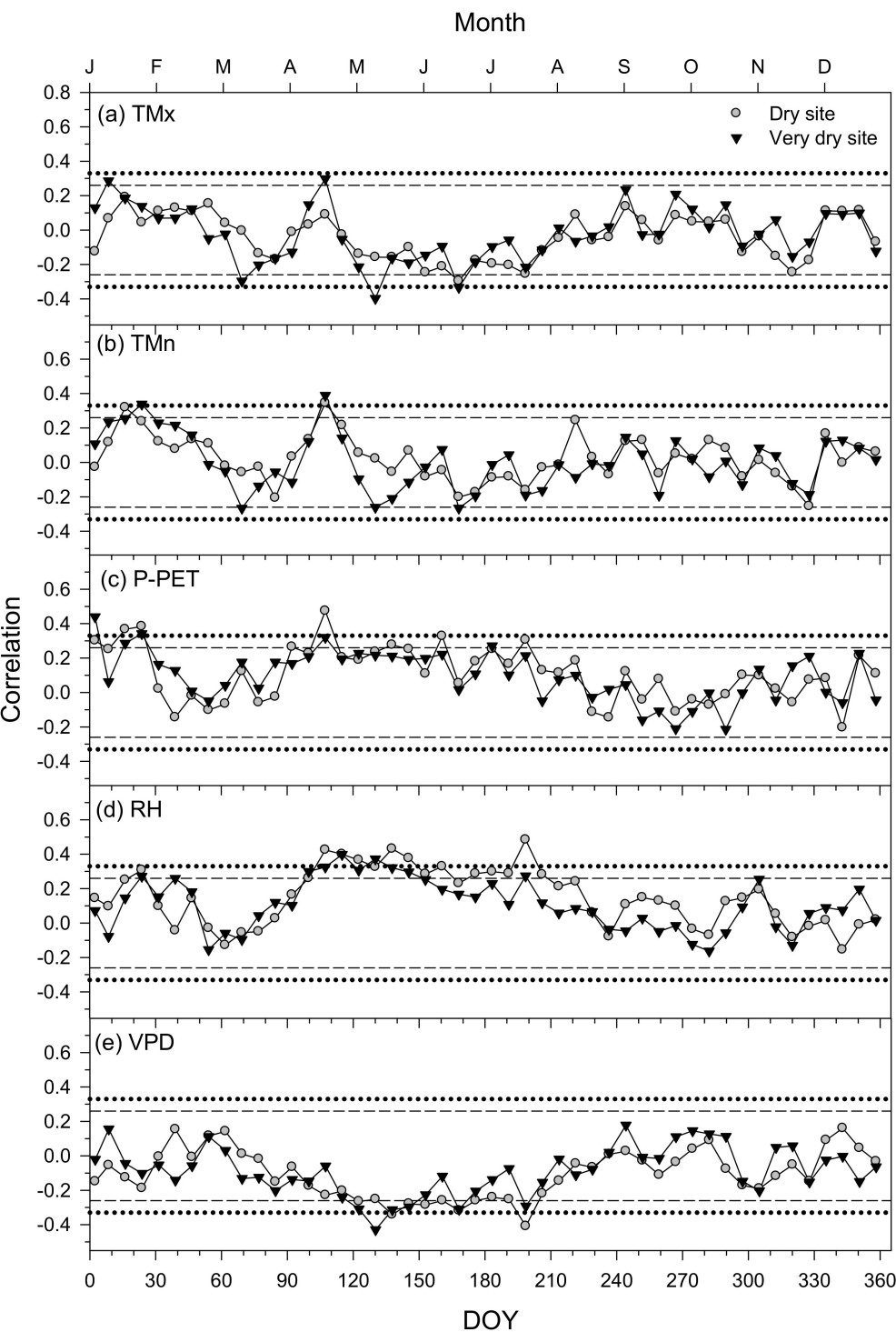

**Figure 9.** The climate–growth relationships of *Juniperus thurifera* were assessed at a weekly resolution at the dry and very dry sites. The analyzed climate variables were (**a**) TMx, (**b**), TMn, (**c**), P-PET, (**d**) RH, and (**e**) VPD. The horizontal dashed and dotted lines indicate the 0.05 and 0.01 significance levels, respectively.

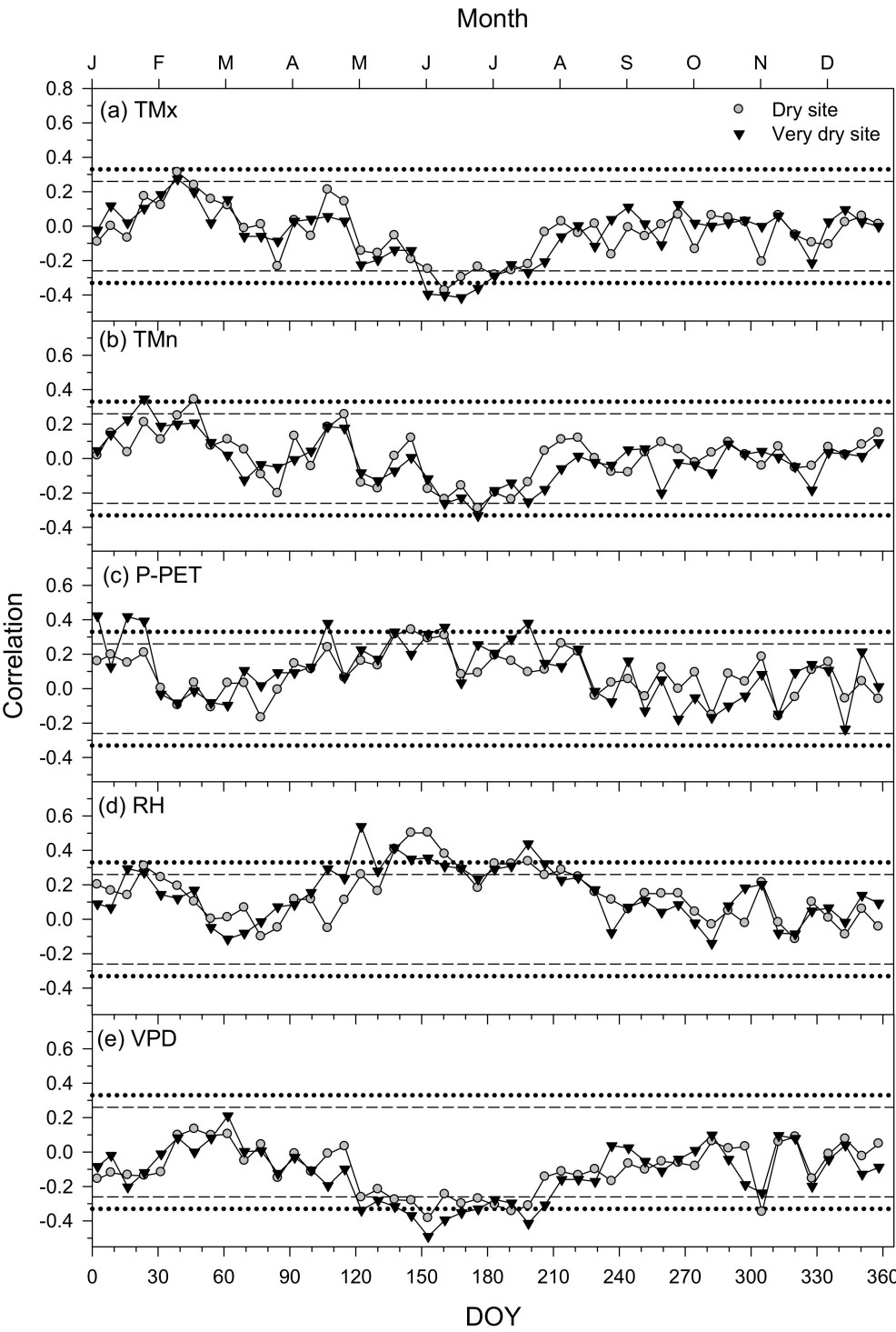

**Figure 10.** The climate–growth relationships of *Juniperus phoenicea* were assessed at a weekly resolution at the dry and very dry sites. The analyzed climate variables were (**a**) TMx, (**b**), TMn, (**c**), P-PET, (**d**) RH, and (**e**) VPD. The horizontal dashed and dotted lines indicate the 0.05 and 0.01 significance levels, respectively.

### 3.5. Climatic Constraints of Growth Inferred from the VS-Lite Model

The VS-Lite model accurately tracked the year-to-year growth variability of the studied trees and shrubs (Table 3). Overall, we found more pronounced soil moisture than temperature limitations on growth (gM < gT) for *P. halepensis* at the very dry site and *Ephedra* at the dry site, with similar responses between junipers at each site (Figure 11). The model showed that the growth of the four species was limited by the low temperature at the beginning and the end of the growing season (gT < gM), with warmer conditions for the *P. halepensis*, and by the lower soil moisture availability during spring, summer, and autumn. The model showed a lower soil moisture peak for *P. halepensis* (from February to late November) at the very dry site and for *Ephedra* (from February to November) at the dry site (see Figure 11).

**Table 3.** The Pearson correlation coefficients (*r*) that were calculated between the mean series of the observed and simulated ring width indices for the calibration period (1960-2019). The Pearson correlation values (*r*) were always significant at the 0.05 level and are presented. The statistics of the Bayesian estimation of the growth response parameters ($T_1$, $T_2$, $M_1$, and $M_2$ for minimum and optimal temperature and soil moisture values, respectively) are also shown. See Table S2 for the cross-validated subperiods.

| Site Type | Species | *r* | $T_1$ (°C) | $T_2$ (°C) | $M_1$ (v/v) | $M_2$ (v/v) |
|---|---|---|---|---|---|---|
| Dry site | *Pinus halepensis* | 0.69 | 3.22 | 11.02 | 0.026 | 0.235 |
| | *Juniperus thurifera* | 0.58 | 8.46 | 20.01 | 0.015 | 0.361 |
| | *Juniperus phoenicea* | 0.49 | 6.66 | 17.62 | 0.094 | 0.146 |
| | *Ephedra nebrodensis* | 0.65 | 7.56 | 10.38 | 0.007 | 0.316 |
| Very dry site | *Pinus halepensis* | 0.78 | 5.72 | 12.76 | 0.096 | 0.352 |
| | *Juniperus thurifera* | 0.58 | 4.69 | 12.80 | 0.014 | 0.289 |
| | *Juniperus phoenicea* | 0.67 | 5.20 | 15.01 | 0.001 | 0.284 |
| | *Ephedra nebrodensis* | 0.38 | 6.17 | 17.39 | 0.003 | 0.235 |

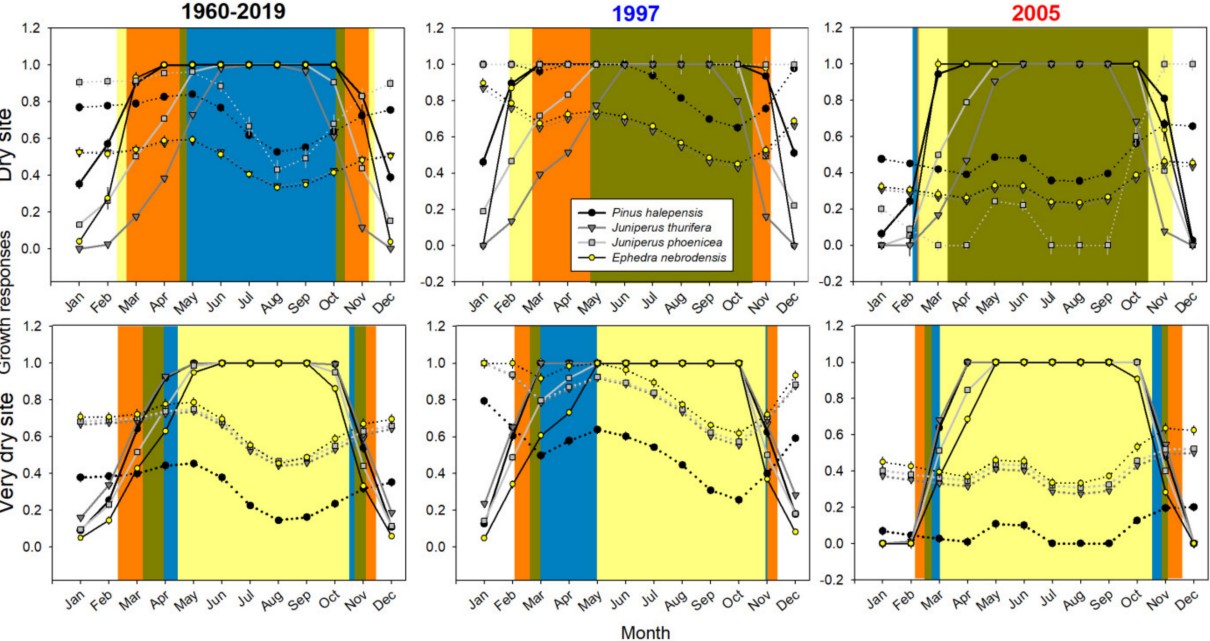

**Figure 11.** Simulated monthly growth response curves (means ± standard errors) of the four study species. Selected wet (1997) and dry (2005) years (see Figures 1 and 3) are also shown. The growth responses consider the temperature (gT, solid lines) and soil moisture parameters (gM, dotted lines). The orange, green, blue, and yellow areas, encompassing late prior winter to late current autumn months, indicate periods that had soil moisture limitations for *P. halepensis*, *J. thurifera*, *J. phoenicea*, and *E. nebrodensis*, respectively.

The modeled growth responses to temperature during wet years were higher at the dry site for trees than in shrubs, whereas the growth responses to the soil moisture during wet years were lower at the very dry site for the *P. halepensis* (Figure 11). The mean growth response (gT, gM) during the dry years indicated longer soil moisture limitations for *Ephedra* at the dry site and for *P. halepensis* at the very dry site, with a lower soil moisture peak for *J. phoenicea* at the dry site and for *P. halepensis* at the very dry site. The highest deviation of the mean growth conditions during the 2005 drought was recorded for *J. phoenicea* at the dry site and for *P. halepensis* at the very dry site (Figure 11). The estimated minimum and optimal thresholds for growth ($T_1$, $T_2$, $M_1$, $M_2$) showed the highest sensitivity of junipers at the dry site to cold temperatures (maximum $T_2$ values) and those of *Ephedra* at the dry site and *P. halepensis* at the very dry site to low soil moisture conditions (maximum $M_2$ values, see Table 3). In addition, *J. thurifera* at both sites and *P. halepensis* at the very dry site showed high soil moisture requirements for optimal growth conditions ($M_2$), confirming the sensitivity of trees to dry and warm conditions during the growing season (Figure 11). Overall, trees showed higher sensitivity to soil moisture than shrubs under increasingly warmer and drier climate conditions (Table S1).

The monthly growth responses to soil moisture (gM) and temperature (gT) increased at the very dry site more than at the dry site since the 1980s, mainly in spring and late summer, due to the lower soil water availability and a significant rise in the spring temperature at the very dry site (Figure 12). This trend toward warmer and drier conditions during the growing season and their negative impact on growth variability was notable for *P. halepensis* from the very dry site, which was the species and site that was more constrained by water deficits. However, the growth responses to temperature were more stable in the dry site. The estimated period with soil moisture limitations (gM < gT) increased faster in trees from the very dry site and for *J. phoenicea* after 1990s at the dry site, with a tendency to present lower soil moisture peaks (the growth was more limited by low soil moisture) during late summer (Figure 12).

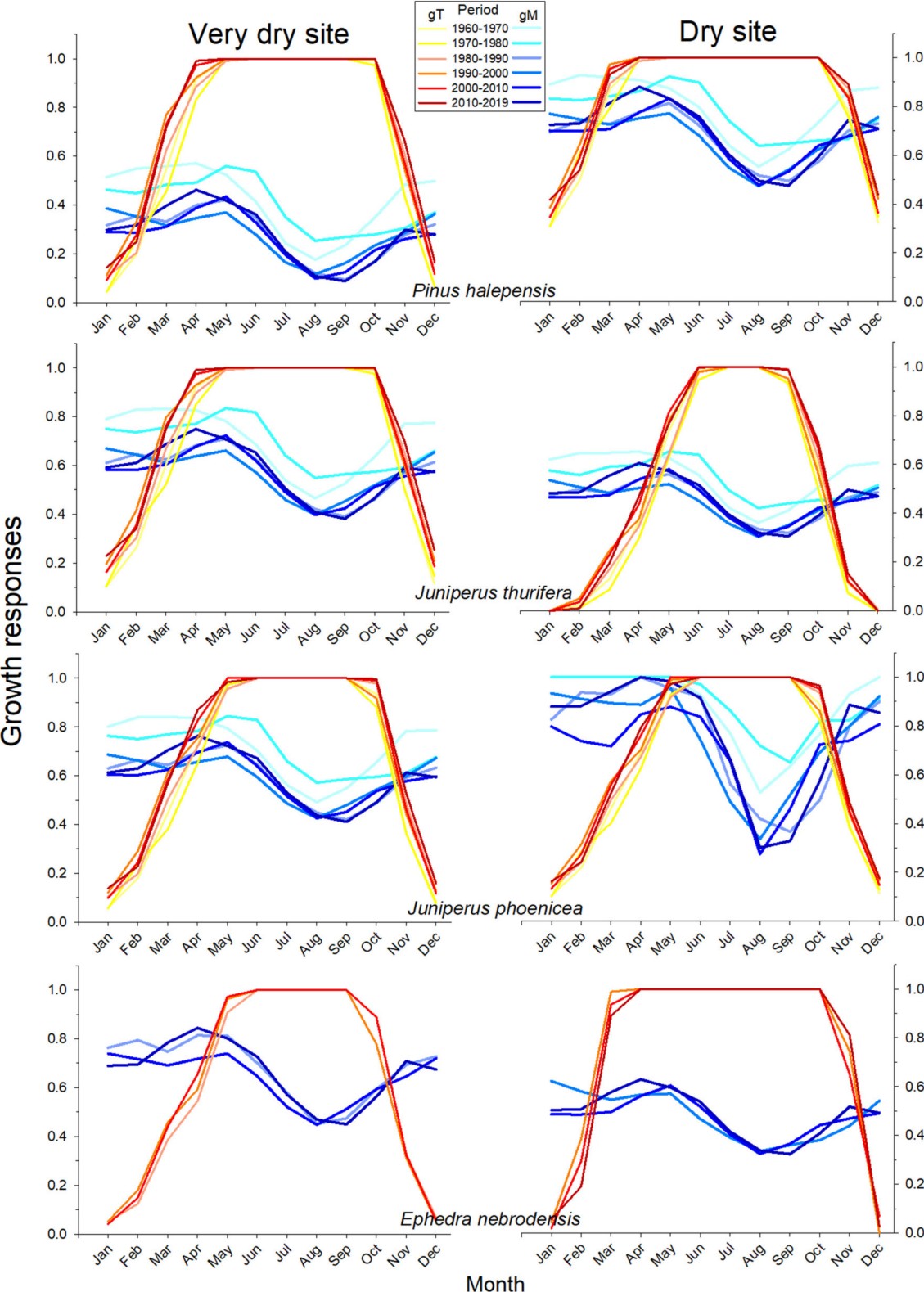

**Figure 12.** Temporal shifts in growth responses to temperature (gT, red to yellow-tone lines) and soil moisture (gM, blue-tone lines) averaged over decades (1960–1970, 1970–1980, 1980–1990, 1990–2000, 2000–2010, 2010–2019) for the four studied species at both sites. The increasingly hotter (higher gT) and drier (lower gM) climatic conditions during spring and autumn are becoming major growth constraints at both sites.

## 4. Discussion

The intra-annual growth pattern was characterized by a major spring peak for all species and a minor autumn peak for some species (pines and junipers), which is consistent with previous xylogenesis studies [7,38,39]. The different phenological patterns of growth were not associated with growth habit, but with phylogeny, since both trees and shrubs showed this bimodal pattern that has already been described in Mediterranean pines and junipers from very dry sites [7,38,39]. The use of basal samples in *Ephedra* could also explain its lower MS and Rbar values and affect the correlations between climate and growth, but assessing the importance of this factor would require taking basal cores in trees, which is out of the scope of this study. The unimodal growth pattern of *Ephedra* at the very dry site and its early growth start (April) suggested a great dependence on prior winter precipitation and early spring soil moisture. This may be explained by the reliance on shallow soil water pools (depth < 50 cm) in xerophytic shrubs with narrow leaves, whilst tree species may be able to explore deeper soil water sources, particularly in the dry summer [40,41]. In the case of the pine and juniper tree and shrub species coexisting in semi-arid regions, junipers tend to use deep soil water sources during wet seasons (spring, autumn), whilst pines are more able to use deeper water pools during the dry summer [42,43], albeit similar short-term responses of the two species have been also observed in response to dry conditions [44,45]. Such varied responses agree with the existence of a dimorphic root system, which allows evergreen trees and shrubs to use shallow soil water sources after rain events and deep sources during dry seasons [46].

The differential use of water resources and ecophysiological strategies (isohydric pine vs. anisohydric junipers) could explain the different climate–growth responses observed in coexisting junipers and pines, regardless of their growth habit. At monthly scales, the growth of the trees *P. halepensis* and *J. thurifera* depended on wet and cool conditions and high soil moisture in winter and spring, whereas the growth of the shrub *J. phoenicea* was improved by cool June–July conditions and elevated soil moisture in July. This was also observed in the analyses at weekly scales, which uncovered the sensitivity of *J. phoenicea* growth to warm and dry (low RH, elevated VPD) June–July conditions, particularly at the very dry site. The positive association of growth with July soil moisture was also observed in *J. thurifera* from the dry site. The dependency of the junipers' growth on summer soil moisture indicates a lower capacity to keep some slow growth in summer as compared with the drought-tolerant and water-saving *P. halepensis* [47]. This would explain the more bimodal growth patterns of junipers as compared with coexisting pines [39] and a high potential to grow in wet autumns, particularly for *J. phoenicea*, which showed the most delayed autumn growth peak. The *Ephedra* positive response to autumn temperatures at the dry site could suggest a bimodal behavior in less stressful sites but the intra-annual growth data do not confirm this. Further research could follow the xylogenesis of this species at sites with different water availability to test that idea.

The sensitivity to warm conditions leading to a high evaporative demand make junipers prone to drought-induced dieback in response to elevated spring–summer temperatures and at sites with soils showing a low water-holding capacity [12,13]. The climate–growth associations of *Ephedra* also make it a candidate species to show dieback and mortality in response to warm and dry summer conditions, given its sensitivity to summer soil moisture, albeit they may also form coarse roots and access moderately deep (10–50 cm) soil water sources [48]. The response to summer climate conditions was also found for *Ephedra procera* growing in Iranian deserts with ≈250 mm of annual precipitation [20], but not in other species growing in cold mountain sites [21]. In contrast, warm and dry winter-to-spring conditions could trigger growth decline and dieback for *P. halepensis* by inducing xylem embolism, particularly in the fine roots [4,49,50].

The VS-Lite results (Figures 11 and 12) agreed with the measured growth rates (Table 2) and, importantly, with the climate correlations (Figures 4–10) but allowed for better identifying the main constraints of growth, such as low soil moisture. The VS-Lite model has been applied to *P. halepensis* and *J. phoenicea* growth data and showed the species'

dependency on winter-to-spring and spring-to-summer cool and wet conditions, respectively [12,37,51]. Our simulations illustrated the pronounced growth sensitivity of the two tree species to dry and warm conditions during the growing season, and were consistent with the higher site-to-site growth variability observed in shrubs. By comparing wet and dry years, the model forecasted a higher sensitivity of *P. halepensis* to increasingly warmer and drier conditions at the very dry site, but also of *J. phoenicea* and *Ephedra* at the dry site (see Figure 12). Moreover, the model detected a rapid shift toward warmer and drier climate conditions and growth constraints due to reduced soil moisture in the 1980s after the wet and cool 1970s. The different responses of trees and shrubs could be interpreted as phenotypic variability and local adaptation of the shrub species to the harsh conditions of the very dry site [39]. A similar explanation could be applied to *J. thurifera* from the very dry site, which showed lower soil moisture limitations than in the dry site. The study *J. thurifera* relict stands could be locally adapted to the harsh climate (dryness) and soil (gypsum) site conditions [39].

We followed a correlative approach by studying the climate–growth associations but it must be considered that the actual drought stress depends on the climate, tree, and stand features, including the genetic composition, site conditions, and soil water dynamics [52,53]. For instance, great intraspecific variability in the soil water uptake [41] and in wood density [54] have been observed in provenance trials of *P. halepensis*, with populations from more arid regions taking up more water from deep soil layers, albeit this was not translated into improved growth. In the field, local factors, such as a higher surface rock cover, may increase the soil water concentration and mitigate the negative impact of warm and dry conditions on the growth and survival of *P. halepensis* at semi-arid sites [55]. Therefore, considering the local site factors (e.g., soil water sources and competition) and individual tree features (e.g., tree size and functional traits, such as wood density) could improve our understanding of the climate–growth associations in dry regions and refine the simulations produced by forward models, such as VS-Lite, which have been mainly applied to trees [37,51]. In the case of shrubs, microsite conditions should be explicitly accounted for [10]. For instance, *Ephedra* species may form longer roots and have higher soil nitrogen concentrations under the canopy than in the interspaces between plants [48].

We argue that a better and more mechanistic approach toward understanding climate–growth responses in tree and shrub species from dry regions, such as semi-arid Mediterranean areas, should integrate functional knowledge with growth sensitivity, including indirect proxies of gas exchange, hydraulics, and soil water uptake, such as wood anatomy and isotopes [12,40,56].

## 5. Conclusions

To conclude, we characterized the intra- and interannual growth patterns of four gymnosperms (the trees *P. halepensis* and *J. thurifera* and the shrubs *J. phoenicea* and *Ephedra*) inhabiting sites with semi-arid Mediterranean conditions. The species showed a major growth peak in spring and a second growth peak in autumn, except for *Ephedra*, which started growing earlier (April) than the other species. Junipers' growth depended on the post-summer soil moisture, confirming a higher capacity to show a second growth peak in autumn (a bimodal pattern), whilst *P. halepensis* growth was more sensitive to climate variability during the prior winter and early spring. The study species were able to tolerate severe and long drought, but very warm and dry conditions reducing the soil moisture and increasing the evaporative demand during winter–spring and summer could lead to growth decline and dieback of *P. halepensis* and *J. phoenicea*. Many shrub species from semi-arid and arid regions, including treeless steppes and deserts, are still understudied and store valuable dendroecological and dendroclimatic information that is still to be discovered.

**Supplementary Materials:** The following are available online at https://www.mdpi.com/1999-4907/12/3/381/s1, Figure S1. Comparison of ring width and its first-order autocorrelation between sites and between growth forms based on the fitted linear mixed-effect models, Figure S2. Comparison

of the year-to-year Ephedra growth (ring-width index) and April to June precipitation variability in the dry site. The selected climate variable showed the highest association with the mean series of ring-width index for the best replicated period showing a correlation of $r = 0.44$ ($p = 0.02$), Figure S3. Comparison of the year-to-year Ephedra growth (RWI, ring width index) and December to January precipitation variability in the very dry site. The selected climate variable showed the highest association with the mean series of ring-width index for the best replicated period showing a correlation of $r = 0.59$ ($p = 0.0009$), Table S1. Model selection for growth rate (RWI, ring width indices) response to 12-month June SPEI between dry and very dry sites (sites) and between shrubs and trees (growth form) considering the period 1989–2019. The coefficients for the 12-month June SPEI (SPEI) and the factors (+ if included) of each proposed linear mixed-effect model are shown together with the degrees of freedom, the change in AIC and the Akaike weight. The selected models are shown in bold, Table S2. Pearson correlation coefficients ($r$) calculated between mean series of observed and simulated ring width indices for the cross-validated sub-periods 1960–1990 and 1991–2019. Pearson correlation coefficients were always significant at the 0.05 level. Statistics of the Bayesian estimation of the growth response parameters (T1, T2, M1, and M2 for minimum and optimal temperature and soil moisture values, respectively) are also shown.

**Author Contributions:** Conceptualization, J.J.C. and R.S.-S.; methodology, C.V., A.G., M.C., R.S.-S., and J.J.C.; software, A.G. and R.S.-S.; validation, C.V., A.G., J.J.C., and R.S.-S.; formal analysis, J.J.C., C.V., A.G., and R.S.-S.; data curation, C.V. and M.C.; writing—original draft preparation, J.J.C., R.S.-S., A.G.; writing—review and editing, all authors; funding acquisition, J.J.C. and R.S.-S. All authors have read and agreed to the published version of the manuscript.

**Funding:** This research was funded by the Spanish Ministry of Science, Innovation, and Universities, grant numbers RTI2018-096884-B-C31 and RTI2018-096884-B-C33. R. Sánchez-Salguero and J.J. Camarero were also supported by the UPO project UPO-1263216, FEDER EU Funds, Andalusia Regional Government, Consejería de Economía, Conocimiento, Empresas y Universidad 2014–2020).

**Institutional Review Board Statement:** Not applicable.

**Informed Consent Statement:** Not applicable.

**Data Availability Statement:** Data will be made available upon reasonable request to the first author.

**Acknowledgments:** We thank Julio Camarero Jiménez for his assistance during the field work and M. Carmen Sancho for preparing the wood cross-sections of the *Ephedra*. We thank the reviewers and the associate editor for revising and improving a previous version of the manuscript.

**Conflicts of Interest:** The authors declare no conflict of interest. The funders had no role in the design of the study; in the collection, analyses, or interpretation of the data; in the writing of the manuscript; in the decision to publish the results.

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
