# Peer review of "Climate Differently Impacts the Growth of Coexisting Trees and Shrubs under Semi-Arid Mediterranean Conditions"

_forests, doi:10.3390/f12030381_

Round 1

Reviewer 1 Report

Dear Authors

The manuscript main aim is to gain knowledge on how different tree and shrub species respond differently to climate conditions in dry or very dry sites. To do so, authors not only perform a classic dendrochronology analyses including monthly climate sensitivity, but also assess intra-annual growth by means of intensive sampling campaigns during a growing season, and also intra-annual growth limitations by means of applying process-based model (Vaganov-Shashkin Lite).

The text is well written and I found sound the applied methods. Author findings are interesting as they enhance the comprehension on a dry ecosystem woody plants growth, relaying not only on tree species, but also in shrubs. Besides, the studied temporal scale (from weeks to decades) together with the accurate climate database (including weekly values) support the robustness of their conclusions. Authors found that similar species growing in the same location can respond in contrasting ways to the same climate conditions, and they provided insights on differential responses in tree compared to shrub species under the same climate conditions. I have some minor comments that hopefully will help authors to improve this manuscript.

First of all, as were not line numbers (at least in the PDF copy I received). In order to avoid confusion countin lines, in each comment I will refer to each section/subsection and the number of line within that section/subsection. I will only count written lines, not spaces between paragraphs or other features, I will not count figures/tables neither figure/table captions for numbering the lines. As an example, I will refer to the line “such as 1997 and 2005, respectively (Figure 3). Other recent dry years with low ring-width” as section 3.3 line 14.

Section 1 Line 24 Bad spelling of Gnetophyta. Besides check for journal rules for Latin scientific names. They might be all italized no matter their taxonomic level.

Section 1 Lines 25-40. I think it is good that the authors give this nice background on Ephedra as it is a not-so-well studied species. However, nothing is said about the other 3 species of the study, and at least two of them are also not so well known (Juniperus thurifera and J. phoenicia). I would suggest to add few lines about the expected or already known behavior of this species on dry habitats, and their possible adaptations as it is done with the Ephedra. I would also suggest to add some hypothesis based on what is said on the introduction about these species. Do authors think that the pine and the two juniper species’ growth will also be constrained by drought? This constrain is expected to be higher/lower in some of the species due to already known anatomical/morphological features? There is some information about rooting systems and vascular feature differences already mentioned in the introduction.

Section 2.1 Lines 2-7. It is not clear whether Monegrillo is considered a dry, a very dry or an intermediate dry site; or if this depends on the species being considered. Table 1 shows that Ephedra was not sampled in the very dry site of Peñaflor, whereas Table 2 shows data of Ephedra in a very dry site (the same occurs with following figures and result and discussion). Were very dry site Ephedra samples taken in Peñaflor?

Figure 1 e, f and g: Not every Y axis is visible in all graphs.

Section 2.2 Lines 4-5. Please specify the height where diameter was taken in shrub species.

Section 2.2 Lines 8-11. It is not clear how the staining was performed. Safranin makes lignified cells become red not blue, so I guess also astrablue or other colorant was used for not lignified cells.

Section 2.2 Lines 11-13 Please specify the software used for digital image analyses, and the country of the Olympus microscope.

Section 2.4 Did authors try to correlate microcore measurements with weekly climatic data? Those correlations might be giving nice information, helping to interpret other results, even if they are based only on one season sampling.

Section 2.5 Lines 12-13 Just a note on the 67% spline that is commonly used: authors might be interested in cheking Klesse 2021 paper discussing about the problems derived by using this detrending method. Anyway, I agree that, in this case, a 67% spline might be the better way to proceed. Stefan Klesse (2021) Critical note on the application of the “two-third” spline. Dendrochronologia 65.

Section 2.6 and last two paragraphs of section 3.5: regarding to the comparison between growth responses of the different species or periods, I think that they would benefit in having some statistical background. An in any case, the statistics will most probably fit with what is visible in the graphs. Maybe a Kolmogorov-Smirnov analysis (or any other) could be used to check for differences in growth responses curves.

Figure 2: I guess there is some mistake in Y axis units (mm day-1) maybe those are micrometers per day.

Section 3.2 Lines 5-7 A linear mixed model accounting for growth habit (tree/shrub) might deal in a better way with each group variance leading to some significant differences. The same applies for following lines.

Section 3.2 Lines 17-19:  1997 and 2005 are used as a wet and dry year respectively in some analyses. I am missing some feeling on how wet or how dry were these years. Maybe a percentile of 1997 and 2005 total precipitation (ore any other statistic) would help to get an idea of the dryness/wetness of those years.

Section 3.2 Line 23: Spelling mistake: site-to-site

Section 3.5 Lines 24-27: I do not fully understand this sentence. I agree that trees showed higher sensitivity to soil moisture than shrubs under the influence of increasingly warmer and drier climate conditions based on M2 parameter: higher M2 values in both trees for the 1991-2019 period compared to the previous one, and also higher M2 values than shrubs. But this is not the case of T2 values: in the dry site, P. halepensis has a lower T2 value during 1991-2019 than in the previous period, and it also has a lower T2 value than J. phoenicia in the last period, and something similar happens in the very dry site.

Figure 11: In dry site graphs, specifically the wet and dry years, Juniperus thurifera growth response to temperature is missing, or at least not visible.

Figure 12: I am wondering why it seems that there is only a difference just between 1960-1980 and 1980-2019 growth responses to soil moisture. It is a sudden change instead of gradual and continuous change as I would expect if it’s only related to warmer and drier conditions in recent times. Do authors have some explanation on this sudden change?

Section 4 Lines 27-32 + Section 5 Lines 5-7: I do not fully understand what authors want to say here. How can a higher capacity of junipers to keep some growth in summer (compared to P. halepensis) can explain its more bimodal growth pattern? If junipers maintain growth during summer the growth pattern would be “less” bimodal than if they don’t maintain it, wouldn’t it?

And I would like to add a suggestion for the discussion. I have the feeling that VS-Lite results can give a bit more information, or at least the information on figures 11 and 12 seem to be “telling the same story” as the climate correlations but in a much clearer and easier way (at least from my point of view). Its easy to see in these figures what seems to be limiting the growth, in which period and for how long for each of the species and sites. Main observable patterns of figures 11 and 12 fit well with tree ring width averages of table 1, and with main climate significant correlations. Just an idea if authors would like to take it, to deal with the climate correlation discussion that its usually a bit tricky and dense.

Author Response

Dear Dr. Camarero,

Thank you for submitting the following manuscript to Forests:

Manuscript ID: forests-1107701
Type of manuscript: Article
Title: Climate differently impacts growth of coexisting trees and shrubs under semi-arid Mediterranean conditions

Authors: J. Julio Camarero *, Cristina Valeriano, Antonio Gazol, Michele Colangelo, Raúl Sánchez-Salguero

Received: 27 January 2021

E-mails: [email protected], [email protected], [email protected], [email protected], [email protected]

Submitted to section: Forest Ecology and Management, https://www.mdpi.com/journal/forests/sections/Ecology_Management
Dendrochronology and Dendroclimatology in the Mediterranean https://www.mdpi.com/journal/forests/special_issues/Dendrochronology_Dendroclimatology

It has been reviewed by experts in the field and we request that you make major revisions before it is processed further. Please find your manuscript and the review reports at the following link: https://susy.mdpi.com/user/manuscripts/resubmit/f36cf725deeba38e3097869d5075d922

Your co-authors can also view this link if they have an account in our submission system using the e-mail address in this message.

Please revise the manuscript according to the reviewers' comments and upload the revised file within 7 days. Use the version of your manuscript found at the above link for your revisions, as the editorial office may have made formatting changes to your original submission. Any revisions should be clearly highlighted, for example using the "Track Changes" function in Microsoft Word, so that changes are easily visible to the editors and reviewers. Please provide a cover letter to explain point-by-point the details of the revisions in the manuscript and your responses to the reviewers' comments. Please include in your rebuttal if you found it impossible to address certain comments. The revised version will be inspected by the editors and reviewers. Please detail the revisions that have been made, citing the line number and exact change, so that the editor can check the changes expeditiously. Simple statements like ‘done’ or ‘revised as requested’ will not be accepted unless the change is simply a typographical error.

Please carefully read the guidelines outlined in the 'Instructions for Authors' on the journal website https://www.mdpi.com/journal/forests/instructions and ensure that your manuscript resubmission adheres to these guidelines. In particular, please ensure that abbreviations have been defined in parentheses the first time they appear in the abstract, main text, and in figure or table captions; citations within the text are in the correct format; references at the end of the text are in the correct format; figures and/or tables are placed at appropriate positions within the text and are of suitable quality; tables are prepared in MS Word table format, not as images; and permission has been obtained and there are no copyright issues.

If the reviewers have suggested that your manuscript should undergo extensive English editing, please have the English in the manuscript thoroughly checked and edited for language and form. Alternatively, MDPI provides an English editing service checking grammar, spelling, punctuation and some improvement of style where necessary for an additional charge (extensive re-writing is not included), see details at https://www.mdpi.com/authors/english.

Do not hesitate to contact us if you have any questions regarding the revision of your manuscript. We look forward to hearing from you soon.

Kind regards, Aleksandar Markovic
Assistant Editor

MDPI, St. Alban-Anlage 66
4052 Basel, Switzerland
Tel.: +41 61 683 77 34, www.mdpi.com

> Dear editor, we thank you and the reviewers for editing and reviewing our ms. Here we provide our point-by-point responses to the editors’ and reviewers’ queries indicating the corresponding sections and line numbers where changes were made in the revised ms.

We thank you for your editorial concern.

Sincerely,

JJ Camarero

************************

Reviewer 1

Dear Authors

The manuscript main aim is to gain knowledge on how different tree and shrub species respond differently to climate conditions in dry or very dry sites. To do so, authors not only perform a classic dendrochronology analyses including monthly climate sensitivity, but also assess intra-annual growth by means of intensive sampling campaigns during a growing season, and also intra-annual growth limitations by means of applying process-based model (Vaganov-Shashkin Lite).

The text is well written and I found sound the applied methods. Author findings are interesting as they enhance the comprehension on a dry ecosystem woody plants growth, relaying not only on tree species, but also in shrubs. Besides, the studied temporal scale (from weeks to decades) together with the accurate climate database (including weekly values) support the robustness of their conclusions. Authors found that similar species growing in the same location can respond in contrasting ways to the same climate conditions, and they provided insights on differential responses in tree compared to shrub species under the same climate conditions. I have some minor comments that hopefully will help authors to improve this manuscript.

> Dear reviewer, we thank you for your positive comments on the ms.

First of all, as were not line numbers (at least in the PDF copy I received). In order to avoid confusion counting lines, in each comment I will refer to each section/subsection and the number of line within that section/subsection. I will only count written lines, not spaces between paragraphs or other features, I will not count figures/tables neither figure/table captions for numbering the lines. As an example, I will refer to the line “such as 1997 and 2005, respectively (Figure 3). Other recent dry years with low ring-width” as section 3.3 line 14.

> We are sorry about the lack of line numbers and we thank you for your efforts indicating the changes along each section. We suspect that the use of the journal template file avoids adding line numbers. We followed your comments section by section.

Section 1 Line 24 Bad spelling of Gnetophyta. Besides check for journal rules for Latin scientific names. They might be all italized no matter their taxonomic level.

> Section, line 24: we corrected the spelling of Gnetophyta and italized the scientific names.

Section 1 Lines 25-40. I think it is good that the authors give this nice background on Ephedra as it is a not-so-well studied species. However, nothing is said about the other 3 species of the study, and at least two of them are also not so well known (Juniperus thurifera and J. phoenicia). I would suggest to add few lines about the expected or already known behavior of this species on dry habitats, and their possible adaptations as it is done with the Ephedra. I would also suggest to add some hypothesis based on what is said on the introduction about these species. Do authors think that the pine and the two juniper species’ growth will also be constrained by drought? This constrain is expected to be higher/lower in some of the species due to already known anatomical/morphological features? There is some information about rooting systems and vascular feature differences already mentioned in the introduction.

> We added two lines to comment on the two juniper species and their potential adaptations to withstand drought as the reviewer suggested. We focused on adaptations related to rooting systems (usually shallow in junipers) and wood anatomy (smaller lumen area) as potential features provide adaptive value to junipers in dry habitats.

Section 2.1 Lines 2-7. It is not clear whether Monegrillo is considered a dry, a very dry or an intermediate dry site; or if this depends on the species being considered. Table 1 shows that Ephedra was not sampled in the very dry site of Peñaflor, whereas Table 2 shows data of Ephedra in a very dry site (the same occurs with following figures and result and discussion). Were very dry site Ephedra samples taken in Peñaflor?

> We clarified these issues in this section. We sampled Ephedra in the very dry site Peñaflor. The Monegrillo site was considered a dry site. The Table 1 was modified accordingly.

Figure 1 e, f and g: Not every Y axis is visible in all graphs.

> We are sorry. We corrected these figures since they share the same Y axes (temperature, precipitation).

Section 2.2 Lines 4-5. Please specify the height where diameter was taken in shrub species.

> We are sorry. This was the basal diameter.

Section 2.2 Lines 8-11. It is not clear how the staining was performed. Safranin makes lignified cells become red not blue, so I guess also astrablue or other colorant was used for not lignified cells.

> We are sorry. We used cresyl violet acetate which reacts with lignin to differently stain lignified and unlignified cells. This has been corrected in the indicated section.

Section 2.2 Lines 11-13 Please specify the software used for digital image analyses, and the country of the Olympus microscope.

> Done, we indicated both.

Section 2.4 Did authors try to correlate microcore measurements with weekly climatic data? Those correlations might be giving nice information, helping to interpret other results, even if they are based only on one season sampling.

> We did not try to calculate those correlations because of the low sample size in some species (e.g., n = 10 meaurements in Ephedra) and because that is part of another study and we considered it out of the scope in this study.

Section 2.5 Lines 12-13 Just a note on the 67% spline that is commonly used: authors might be interested in cheking Klesse 2021 paper discussing about the problems derived by using this detrending method. Anyway, I agree that, in this case, a 67% spline might be the better way to proceed. Stefan Klesse (2021) Critical note on the application of the “two-third” spline. Dendrochronologia 65.

> We thank you for the comment and we will read the commented reference.

Section 2.6 and last two paragraphs of section 3.5: regarding to the comparison between growth responses of the different species or periods, I think that they would benefit in having some statistical background. An in any case, the statistics will most probably fit with what is visible in the graphs. Maybe a Kolmogorov-Smirnov analysis (or any other) could be used to check for differences in growth responses curves.

> Following your comments and the reviewer 2 queries we calculated these comparisons and fitted linear mixed-effects models to consider the whole individual variability in growth variables and to compare growth forms (trees vs. shrubs) and habitat types (dry vs. very dry sites). These analyses have been added in the revised ms.

Figure 2: I guess there is some mistake in Y axis units (mm day-1) maybe those are micrometers per day.

> We are very sorry. The figure was showing the rate of production of new cells. We converted it now the radial-growth rates and changed the text accordingly.

Section 3.2 Lines 5-7 A linear mixed model accounting for growth habit (tree/shrub) might deal in a better way with each group variance leading to some significant differences. The same applies for following lines.

> We agree and we preformed such analyses as suggested. We added the new methods and the related results to the revised ms.

Section 3.2 Lines 17-19:  1997 and 2005 are used as a wet and dry year respectively in some analyses. I am missing some feeling on how wet or how dry were these years. Maybe a percentile of 1997 and 2005 total precipitation (ore any other statistic) would help to get an idea of the dryness/wetness of those years.

> We agree and we addes such information to the referred section.

Section 3.2 Line 23: Spelling mistake: site-to-site

> We corrected the spelling as indicated.

Section 3.5 Lines 24-27: I do not fully understand this sentence. I agree that trees showed higher sensitivity to soil moisture than shrubs under the influence of increasingly warmer and drier climate conditions based on M2 parameter: higher M2 values in both trees for the 1991-2019 period compared to the previous one, and also higher M2 values than shrubs. But this is not the case of T2 values: in the dry site, P. halepensis has a lower T2 value during 1991-2019 than in the previous period, and it also has a lower T2 value than J. phoenicia in the last period, and something similar happens in the very dry site.

> We agree with your comment and rephrased the sentence by removing the reference to the T2 parameter.

Figure 11: In dry site graphs, specifically the wet and dry years, Juniperus thurifera growth response to temperature is missing, or at least not visible.

> We thank you for the comment and we have revised and amended the figure.

Figure 12: I am wondering why it seems that there is only a difference just between 1960-1980 and 1980-2019 growth responses to soil moisture. It is a sudden change instead of gradual and continuous change as I would expect if it’s only related to warmer and drier conditions in recent times. Do authors have some explanation on this sudden change?

> We thank you for the insight. In fact, there was a rapid shift towards drier climate conditions in the 1980s because the 1970s were wet and cool in the study area,whereas the 1980s, 1990s and 2000-2010s were increasingly warmer and drier. We added this comment to the revised ms.

Section 4 Lines 27-32 + Section 5 Lines 5-7: I do not fully understand what authors want to say here. How can a higher capacity of junipers to keep some growth in summer (compared to P. halepensis) can explain its more bimodal growth pattern? If junipers maintain growth during summer the growth pattern would be “less” bimodal than if they don’t maintain it, wouldn’t it?

> You are right and there is a misunderstanding here. It is P. halepensis which is better able to maintain some growth during the dry summer period. We corrected it in the revised ms.

And I would like to add a suggestion for the discussion. I have the feeling that VS-Lite results can give a bit more information, or at least the information on figures 11 and 12 seem to be “telling the same story” as the climate correlations but in a much clearer and easier way (at least from my point of view). Its easy to see in these figures what seems to be limiting the growth, in which period and for how long for each of the species and sites. Main observable patterns of figures 11 and 12 fit well with tree ring width averages of table 1, and with main climate significant correlations. Just an idea if authors would like to take it, to deal with the climate correlation discussion that its usually a bit tricky and dense.

> We thank you for such useful comment. We have tried to implement it in the revised discussion by playing more attention the VS-Lite figures. Thanks again for such useful revision.

Reviewer 2 Report

The reviewed work is very extensive in terms of the presented content. It comprehensively shows methods of analyzing the influence of climatic factors on the growth of annual rings. In my opinion, it is the main value of the paper.

The most important notes for the paper are presented below. Their order does not show their importance. 

p.2 l. 27: Probably mistake: "semi-aird"

Chapter 2.2: Most of the samples were taken from 1.3 m and in the case of Ephedra samples were taken from the plant base. In my opinion, it needs an explanation due to the possible additional influence of mechanical factors on the formation of tree-rings. It could be a reason for lower MS and Rbar values and affect correlation with climate factors.

Chapter 2.4: Why data from the climatic stations were not used? Was the comparison of the data from the Climate Explorer with the data from the station performed? 

Figure 1. On climate diagrams, Y-axis labels are "cut off". If the precipitation data series will be shown as a bar-plot it would make the graph easier to read.

Chapter 3.2: The t-test is not the best statistical method if you want to analyze the mean ring-width differences between plant species and sites. In this case, the analysis of variance could be the more suitable method.

Table 2. The table is too wide for the page.

Based on figure 3 I have two questions: Do the missing-rings were observed and how frequent they were? How stable in time is the climate-growth relation?

p. 9-10. The font size is changed.

Figures 4, 5, 6 due to different plot types and different vertical axes scaling, are not easy to read. Both vertical axes show the correlation coefficient values, so what's the point to make the plots so complicated. Another thing is the use of a line-linked point plot for the soil moisture series. The lines suggest the continuity of the relationship, but the horizontal axis presents two types of time periods: months and seasons. My suggestion: unify the scales and use a bar-chart for all data series.

Chapter 3.3 and 3,4. I have many doubts about the assessment of the climate-growth relationship. For example: based on Figure 5, in described case of P. halapensis I see a relationship between radial growth and soil moisture in the winter (how it is stated), but for in the June-July period, the values of the correlation coefficient are small and not significant.  In the description of figure 8 you stated "cool and wet (high P-PET)
January conditions also improved growth in the very dry site", but there is a positive (but not significant) correlation with temperature. So, not "cool and wet" but only wet January is correlated with radial growth.

In many cases, the descriptions suggest the existence of a cause-and-effect relationship, while the results show only a correlation.

Chapter 3.5. Based on references 32 and 33 i have some doubts about the parameterization of the model, but I have no possibility to check it. Could you describe in more detail this part of your work?

Figure 11. Description: ..." four study Scheme 1960".

Figure 12 is hard to read and it is hard to find the temporal shifts in growth responses over the following decades.

Author Response

Reviewer 2

The reviewed work is very extensive in terms of the presented content. It comprehensively shows methods of analyzing the influence of climatic factors on the growth of annual rings. In my opinion, it is the main value of the paper.

> We thank you for your positive comments.

The most important notes for the paper are presented below. Their order does not show their importance. 

p.2 l. 27: Probably mistake: "semi-aird"

> We corrected the text.

Chapter 2.2: Most of the samples were taken from 1.3 m and in the case of Ephedra samples were taken from the plant base. In my opinion, it needs an explanation due to the possible additional influence of mechanical factors on the formation of tree-rings. It could be a reason for lower MS and Rbar values and affect correlation with climate factors.

> Sampled Ephedra individuals are low shrubs so we had to take basal cross sections. We agree this could affect MS and Rbar or climate-growth correlations, but quantifying that effect is difficult since we did not take basal samples in trees and it is out of the scope of this study. Nevertheless, we added a comment on this issue in the revised Discussion.

Chapter 2.4: Why data from the climatic stations were not used? Was the comparison of the data from the Climate Explorer with the data from the station performed? 

> We used interpolated, 0.1º-gridded climate data because there are no stations with long and homogeneous records near the study sites. The comparison with local climate data was not performed given the different spatial scales and temporal resolutions and the lack of reliable local climate data during the last 30 years in both study sites.

Figure 1. On climate diagrams, Y-axis labels are "cut off". If the precipitation data series will be shown as a bar-plot it would make the graph easier to read.

> Sorry, we showed now the complete y axes. The use of lines for temperature and precipitation is clear enough and allows readers to identify the drought period.

Chapter 3.2: The t-test is not the best statistical method if you want to analyze the mean ring-width differences between plant species and sites. In this case, the analysis of variance could be the more suitable method.

> Sorry, we provided new analyses including linear mixed-effects models.

Table 2. The table is too wide for the page.

> We re-arranged the table to fit it in the page.

Based on figure 3 I have two questions: Do the missing-rings were observed and how frequent they were? How stable in time is the climate-growth relation?
> We did not observe missing rings. We did not assess the stability of climate-growth relationships because the comparison with short shrub series (e.g., Ephedra) does not allow performing this assessment in all species. The period common to these analyses is 30-year long which is too short for assessing the stability of climate-growth associations.

  1. 9-10. The font size is changed.

> We corrected the font size.

Figures 4, 5, 6 due to different plot types and different vertical axes scaling, are not easy to read. Both vertical axes show the correlation coefficient values, so what's the point to make the plots so complicated. Another thing is the use of a line-linked point plot for the soil moisture series. The lines suggest the continuity of the relationship, but the horizontal axis presents two types of time periods: months and seasons. My suggestion: unify the scales and use a bar-chart for all data series.

> The issue here is the use of climate data series of different length since the series of estimated soil moisture is shorter than the others. This explains why soil moisture (blue points) has different significance levels (horizontal dotted blue lines).

The figure is based on monthly and seasonal climate data. Note also that this type of plot is usually employed in similar dendroecological studies reporting climate-growth correlations.

Chapter 3.3 and 3,4. I have many doubts about the assessment of the climate-growth relationship. For example: based on Figure 5, in described case of P. halapensis I see a relationship between radial growth and soil moisture in the winter (how it is stated), but for in the June-July period, the values of the correlation coefficient are small and not significant.  In the description of figure 8 you stated "cool and wet (high P-PET).

> Our analyses are based on perfectly cross-dated and measured wood series. Interpolated climate or soil moisture data may have some uncertainty as there may be some measurement errors. However, we are very confident on these findings. For instance, in the study area wet and cool conditions in the prior winter (leading to high soil moisture) are usually associated to high growth of P. halepensis, which usually peaks from April to June. This is explained because those conditions allow recharging soil moisture pools and this higher soil moisture enhances spring growth. This is a clear example of lagged growth response to climate.

January conditions also improved growth in the very dry site", but there is a positive (but not significant) correlation with temperature. So, not "cool and wet" but only wet January is correlated with radial growth.

> OK, we rephrase the comments stating “cool and wet” and changed them by “wet” conditions.

In many cases, the descriptions suggest the existence of a cause-and-effect relationship, while the results show only a correlation.

> We follow a correlative approach so we can only infer cause-effect relationships, but we are only describing correlations. We toned down any expression suggesting such cause-effect relationship since we are aware that our analyses preclude achieving those conclusions.

Chapter 3.5. Based on references 32 and 33 i have some doubts about the parameterization of the model, but I have no possibility to check it. Could you describe in more detail this part of your work?

> We do not understand your concerns. The correlations presented in Tables 3 and S2 indicate that the parameterization of the model adequately allowed reproducing growth rates of the study species. We followed a similar approach as Tolwinski-Ward et al. and used as input monthly climate data and observed growth rates (RWI). The VS-Lite model just uses two temperature and two soil moisture parameters which are provided in the referred Tables. The fits of the model (predicted RWI series) were significantly correlated with the observed species RWI series.

Figure 11. Description: ..." four study Scheme 1960".

> We corrected the legend. Thank you.

Figure 12 is hard to read and it is hard to find the temporal shifts in growth responses over the following decades.

> We think the figure clearly shows a shift towards drier conditions in the 1980s since the 1970s were wet. This remark was done by reviewer 1 who noted that the VS-Lite allowed delimiting the main climatic constraints (e.g., low soil moisture) of the study sites and species.

Editor comments:

1) novelty of the content

-An important analysis, adds to the global data base, but not novel; there are about 40 papers on this topic in the last 10 yrs.
> We understand your comment, but our team is mainly focused on drought impacts on radial growth of woody plant species. Note that the studies comparing coexisting tree and shrub species are rare, so this adds novelty to our approach.

2) potential impact of the manuscript in the relevant field of research

- Because of the interest in droughts vs. hot droughts on carbon acquisition (depressed w both temperature and drought stress), allocation (radial growth as well as all other tissues), and partitioning (respiratory losses), the contribution would be (more) useful if a drought index or indices were used instead of temperature and precipitation separately. A mixed effects, linear model should be used to test the response of the species to the drought index/indices. The response of each species to the drought index, compared between the two sites, one drier than the other, would then be compared statistically.

> We have carried out similar studies using drought indices (e.g., SPEI, PDSI) as you suggest. However, we were more interested on the direct impacts of climate variables with functional or ecophysiological meaning (e.g., water balance, soil moisture) on radial growth. The use of drought indices has other pros, but also has cons because they are relative metrics of drought severity and do not allow these functional approximations. Nevertheless, we also introduced analyses based on the SPEI drought index following your comment. As you suggested we fitted linear mixed-effects models to consider the whole range of individual growth variability and growth responses to climate and drought.

-There are no measures of how physiologically 'stressed' the trees are in either site. Are there any features other than how much carbon was allocated to the tree trunk that the trees were more or less stressed? Some of the tree species show greater diameters and height growth in the 'drier' site. Are they older? If they are stressed, why are the trees taller in the 'drier' site (all species I think except one of the Juniperus species). This needs to be cleared up...

> We just measured radial growth rates as proxies of carbon allocation to wood formation. This is our main measure of “climatic stress” since ring width is a very sensitive variable in response to changes in water availabiluty.

Regarding the size and age differences you commented (cf. Table 2), most variables (diameter, height, age) were quite similar (there were no significant differences in size or age) when comparing the four study species in the dry vs. the very dry sites. However, the mean radial growth rates were lower in the very dry site (0.79 mm) when compared with the dry site (0.94 mm) confirming that the first site was climatically more stressful than the second site.

Kind regards,

Aleksandar Markovic

Assistant Editor

> We thank you for your editorial concern.
